# Autoencoding Random Forests

**Binh Duc Vu**[1]*
binh.vu@kcl.ac.uk

**Jan Kapar**[2,3]*
kapar@leibniz-bips.de

**Marvin N. Wright**[2,3]
wright@leibniz-bips.de

**David S. Watson**[1]
david.watson@kcl.ac.uk

[1]King's College London
[2]Leibniz Institute for Prevention Research and Epidemiology – BIPS
[3]University of Bremen

## Abstract

We propose a principled method for autoencoding with random forests. Our strategy builds on foundational results from nonparametric statistics and spectral graph theory to learn a low-dimensional embedding of the model that optimally represents relationships in the data. We provide exact and approximate solutions to the decoding problem via constrained optimization, split relabeling, and nearest neighbors regression. These methods effectively invert the compression pipeline, establishing a map from the embedding space back to the input space using splits learned by the ensemble's constituent trees. The resulting decoders are universally consistent under common regularity assumptions. The procedure works with supervised or unsupervised models, providing a window into conditional or joint distributions. We demonstrate various applications of this autoencoder, including powerful new tools for visualization, compression, clustering, and denoising. Experiments illustrate the ease and utility of our method in a wide range of settings, including tabular, image, and genomic data.

## 1 Introduction

Engineering compact, informative representations is central to many learning tasks [58, 96, 11, 44, 81]. In supervised applications, it can simplify regression or classification objectives, helping users better understand the internal operations of large, complicated models [42, 104]. In reinforcement learning, embeddings help agents navigate complex environments, imposing useful structure on a potentially high-dimensional state space [2, 63]. In unsupervised settings, latent projections can be used for data compression [69], visualization [93], clustering [97], and generative modeling [92].

The current state of the art in representation learning is dominated by deep neural networks (DNNs). Indeed, the tendency of these algorithms to learn rich embeddings is widely cited as a key component of their success [44], with some even arguing that large language models are essentially compression engines [29]. It is less obvious how to infer latent factors from tree-based ensembles such as random forests (RFs) [17], a popular and flexible function class widely used in areas like bioinformatics [23] and econometrics [3]. DNNs are known to struggle in tabular settings with mixed continuous and categorical covariates, where tree-based ensembles typically match or surpass their performance [86, 48]. Though several authors have proposed methods for computing nonlinear embeddings with RFs (see Sect. 2), these approaches tend to be heuristic in nature. Moreover, the task of decoding latent vectors to recover input data in these pipelines remains unresolved.

---

*Equal contribution.

39th Conference on Neural Information Processing Systems (NeurIPS 2025).

We propose a novel, principled method for autoencoding with RFs. Our primary contributions are: (1) We prove several important properties of the adaptive RF kernel, including that it is asymptotically universal. (2) These results motivate the use of diffusion maps to perform nonlinear dimensionality reduction and manifold learning with RFs. Resulting embeddings can be used for various downstream tasks. (3) We introduce and study multiple methods for decoding spectral embeddings back into the original input space, including exact and approximate solutions based on constrained optimization, split relabeling, and nearest neighbors regression. (4) We apply these methods in a series of experiments and benchmark against a wide array of neural and tree-based alternatives. Our results demonstrate that the RF autoencoder is competitive with the state of the art across a range of tasks including data visualization, compression, clustering, and denoising.

The remainder of this paper is structured as follows. After a review of background material and related work (Sect. 2), we propose and study methods for encoding (Sect. 3) and decoding (Sect. 4) data with RFs. Performance is illustrated in a series of experiments (Sect. 5). Following a brief discussion (Sect. 6), we conclude with directions for future work (Sect. 7).

## 2 Background

Our starting point is the well established connection between RFs and kernel methods [16, 28, 83]. The basic insight is that classification and regression trees (CART) [18], which serve as basis functions for many popular ensemble methods, are a kind of adaptive nearest neighbors algorithm [66]. At the root of the tree, all samples are connected. Each split severs the link between one subset of the data and another (i.e., samples routed to left vs. right child nodes), resulting in a gradually sparser graph as depth increases. At completion, a sample's "neighbors" are just those datapoints that are routed to the same leaf. Given some feature space $\mathcal{X} \subset \mathbb{R}^{d_\mathcal{X}}$, the implicit kernel of tree $b$, $k^{(b)} : \mathcal{X} \times \mathcal{X} \mapsto \{0, 1\}$, is an indicator function that evaluates to 1 for all and only neighboring sample pairs.[1] This base kernel can be used to define different ensemble kernels. For instance, taking an average over $B > 1$ trees, we get a kernel with a simple interpretation as the proportion of trees in which two samples colocate: $k^S(\mathbf{x}, \mathbf{x}') = B^{-1} \sum_{b=1}^{B} k^{(b)}(\mathbf{x}, \mathbf{x}')$. We call this the *Scornet kernel* after one of its noted proponents [83], who showed that $k^S$ is provably close in expectation to the RF similarity function:

$$k_n^{RF}(\mathbf{x}, \mathbf{x}') = \frac{1}{B} \sum_{b=1}^{B} \left( \frac{k^{(b)}(\mathbf{x}, \mathbf{x}')}{\sum_{i=1}^{n} k^{(b)}(\mathbf{x}, \mathbf{x}_i)} \right), \tag{1}$$

where $i \in [n] := \{1, \dots, n\}$ indexes the training samples. This represents the average of *normalized* tree kernels, and fully encodes the information learned by the RF $f_n$ via the identity:

$$f_n(\mathbf{x}) = \sum_{i=1}^{n} k_n^{RF}(\mathbf{x}, \mathbf{x}_i) \, y_i, \tag{2}$$

which holds uniformly for all $\mathbf{x} \in \mathcal{X}$. Though $k^S$ is sometimes referred to as "the random forest kernel" [28, 76], this nomenclature is misleading—only $k_n^{RF}$ satisfies Eq. 2. Non-adaptive variants of $k_n^{RF}$ have been previously studied [109], but we derive several novel properties of this similarity function without any such constraints.

Several nonlinear dimensionality reduction techniques are based on kernels, most notably kernel principal component analysis (KPCA) [80]. We focus in particular on diffusion maps [25, 26], which can be interpreted as a form of KPCA [52]. Bengio et al. [10] establish deep links between these algorithms and several related projection methods, demonstrating how to embed test data in all settings via the Nyström formula, a strategy we adopt below. Inverting any KPCA algorithm to map latent vectors to the input space is a nontrivial task that must be tailored to each specific kernel. For an example with Gaussian kernels, see [72].

Previous authors have explored feature engineering with RFs. Shi and Horvath [85] perform multi-dimensional scaling on a dissimilarity matrix extracted from supervised and unsupervised forests. However, they do not explore the connections between this approach and kernel methods, nor do they

---

[1]This ignores certain subtleties that arise when trees are grown on bootstrap samples, in which case $k^{(b)}$ may occasionally evaluate to larger integers. For present purposes, we presume that trees are grown on data subsamples; see Appx. A.

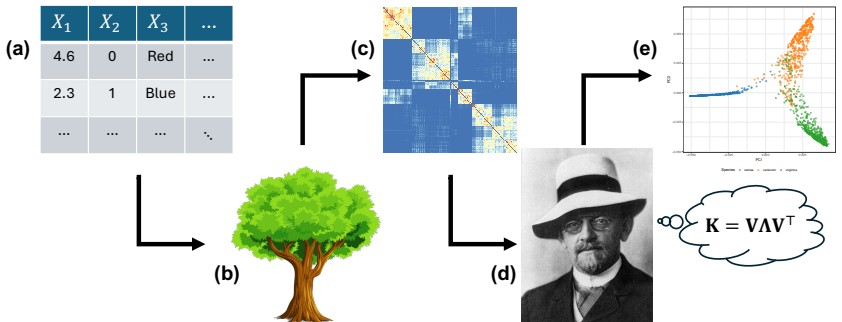

Figure 1: Visual summary of the encoding pipeline. (a) Input data can be a mix of continuous, ordinal, and/or categorical variables. (b) A RF (supervised or unsupervised) is trained on the data. (c) A kernel matrix $\mathbf{K} \in [0,1]^{n \times n}$ is extracted from the ensemble. (d) $\mathbf{K}$ is decomposed into its eigenvectors and eigenvalues, as originally proposed by David Hilbert (pictured). (e) Data is projected onto the top $d_{\mathcal{Z}} < n$ principal components of the diffusion map, resulting in a new embedding $\mathbf{Z} \in \mathbb{R}^{n \times d_{\mathcal{Z}}}$.

propose any strategy for decoding latent representations. More heuristic approaches involve running PCA on a weighted matrix of all forest nodes (not just leaves), a method that works well in some experiments but comes with no formal guarantees [77].

Several generative algorithms based on RFs have been proposed, building on the probabilistic circuit literature [27, 99]. These do not involve any explicit encoding step, although the models they train could be passed through our pipeline to extract latent embeddings (see Sect. 5). Existing methods for sum-product network encoding could in principle be applied to an RF following compilation into a corresponding circuit [95]. However, these can often *increase* rather than *decrease* dimensionality, and do not come with any associated decoding procedure.

Recent work has explored autoencoding for tree- and forest-based models. Carreira-Perpiñán and Gazizov [20] introduce a sparse oblique tree where each node performs a local linear reconstruction via PCA, yielding a tree-structured autoencoder without use of kernels or ensembles. Aumon et al. [4] propose an RF-informed neural autoencoder, which extracts forest-induced sample similarities to inject supervised signals into network embeddings for data visualization. By contrast, we learn latent representations directly from the RF and implement explicit decoding rules for both supervised and unsupervised settings without an auxiliary model.

Perhaps the most similar method to ours, in motivation if not in execution, is Feng and Zhou's encoder forest (eForest) [34]. This algorithm maps each point to a hyperrectangle defined by the intersection of all leaves to which the sample is routed. Decoding is then achieved by taking some representative value for each feature in the subregion (e.g., the median). Notably, this approach does *not* include any dimensionality reduction. On the contrary, the embedding space requires minima and maxima for all input variables, resulting in a representation with double the number of features as the inputs. Working from the conjecture that optimal prediction is equivalent to optimal compression [78, 58, 49], we aim to represent the information learned by the RF in relatively few dimensions.

## 3 Encoding

As a preliminary motivation, we prove that the RF similarity function is a proper kernel with several notable properties. The following definitions are standard in the literature. Let $\mathcal{X}$ be a compact metric space, and let $C(\mathcal{X})$ be the set of all real-valued continuous functions on $\mathcal{X}$.

**Definition 3.1** (Positive semidefinite). A symmetric function $k : \mathcal{X} \times \mathcal{X} \mapsto \mathbb{R}$ is *positive semidefinite* (PSD) if, for all $\mathbf{x}_1, \ldots, \mathbf{x}_n \in \mathcal{X}$, $n \in \mathbb{N}$, and $c_i, c_j \in \mathbb{R}$, we have: $\sum_{i,j}^{n} c_i \, c_j \, k(\mathbf{x}_i, \mathbf{x}_j) \geq 0$.

The Moore-Aronszajn theorem [1] states that PSD kernels admit a unique reproducing kernel Hilbert space (RKHS) [12], providing a rich mathematical language for analyzing their behavior.

**Definition 3.2** (Universal). We say that the RKHS $\mathcal{H}$ is *universal* if the associated kernel $k$ is dense in $C(\mathcal{X})$ with respect to the uniform norm. That is, for any $f^* \in C(\mathcal{X})$ and $\epsilon > 0$, there exists some $f \in \mathcal{H}$ such that $\|f^* - f\|_\infty < \epsilon$.

Several variants of universality exist with slightly different conditions on $\mathcal{X}$ [87]. Examples of universal kernels include the Gaussian and Laplace kernels [88].

**Definition 3.3** (Characteristic). The bounded measurable kernel $k$ is *characteristic* if the function $\mu \mapsto \int_{\mathcal{X}} k(\cdot, \mathbf{x}) \, d\mu(\mathbf{x})$ is injective, where $\mu$ is a Borel probability measure on $\mathcal{X}$.

Characteristic kernels are especially useful for statistical testing, and have inspired flexible nonparametric methods for evaluating marginal and conditional independence [46, 39]. For instance, Gretton et al. [47] show that, when using a characteristic kernel $k$, the maximum mean discrepancy (MMD) between two measures $\mu, \nu$ on $\mathcal{X}$ is zero iff $\mu = \nu$. The MMD is defined as:

$$\text{MMD}^2(\mu, \nu; k) := \mathbb{E}_{\mathbf{x}, \mathbf{x}' \sim \mu}[k(\mathbf{x}, \mathbf{x}')] - 2\mathbb{E}_{\mathbf{x} \sim \mu, \mathbf{y} \sim \nu}[k(\mathbf{x}, \mathbf{y})] + \mathbb{E}_{\mathbf{y}, \mathbf{y}' \sim \nu}[k(\mathbf{y}, \mathbf{y}')].$$

With these definitions in place, we state our first result (all proofs in Appx. A).

**Theorem 3.4** (RF kernel properties). *Assume standard RF regularity conditions (see Appx. A). Then:*
*(a) For all $n \in \mathbb{N}$, the function $k_n^{RF}$ is PSD and the kernel matrix $\mathbf{K} \in [0,1]^{n \times n}$ is doubly stochastic.*
*(b) Let $\{f_n\}$ be a sequence of RFs. Then the associated RKHS sequence $\{\mathcal{H}_n\}$ is asymptotically universal. That is, for any $f^* \in C(\mathcal{X})$ and $\epsilon > 0$, we have:*

$$\lim_{n \to \infty} P\big(\|f^* - f_n\|_{\infty} \geq \epsilon\big) = 0.$$

*(c) The RKHS sequence $\{\mathcal{H}_n\}$ is asymptotically characteristic. That is, for any $\epsilon > 0$, the Borel measures $\mu, \nu$ are equal if and only if:*

$$\lim_{n \to \infty} P\big(\text{MMD}(\mu, \nu; k_n^{RF}) \geq \epsilon\big) = 0.$$

The literature on kernel methods is largely focused on fixed kernels such as the radial basis function (RBF). Among adaptive partitioning alternatives, the Scornet kernel been studied in some detail [28, 83, 76], as has the Mondrian kernel [64, 73, 21]. Zhou and Hooker [109] show that a non-adaptive variant of $k_n^{RF}$ is PSD and bounded on the unit interval (a strictly weaker property than double stochasticity). To the best of our knowledge, we are the first to establish items (b) and (c) for the RF kernel. Thm. 3.4 confirms that $k_n^{RF}$ is flexible, informative, and generally "well-behaved" in ways that will prove helpful for autoencoding.

**Spectral Graph Theory** A key insight from spectral graph theory is that for any PSD kernel $k$, there exists an encoding $g : \mathcal{X} \mapsto \mathcal{Z}$ for any embedding dimension $d_{\mathcal{Z}} < n$ that optimally represents the data, in a sense to be made precise below. This motivates our use of diffusion maps [26, 25], which are closely related to Laplacian eigenmaps [8, 9], an essential preprocessing step in popular spectral clustering algorithms [75, 97, 59]. These methods are typically used with fixed kernels such as the RBF; by contrast, we use the adaptive RF kernel, which is better suited to mixed tabular data.

The procedure begins with a dataset of paired feature vectors $\mathbf{x} \in \mathcal{X} \subset \mathbb{R}^{d_{\mathcal{X}}}$ and outcomes $y \in \mathcal{Y} \subset \mathbb{R}$ sampled from the joint distribution $P_{XY}$.[2] A RF of $B$ trees $f_n$ is trained on $\{\mathbf{x}_i, y_i\}_{i=1}^n$. Using Eq. 1, we construct the kernel matrix $\mathbf{K} \in [0,1]^{n \times n}$ with entries $k_{ij} = k_n^{RF}(\mathbf{x}_i, \mathbf{x}_j)$. This defines a weighted, undirected graph $\mathcal{G}_n$ over the training data. As $\mathbf{K}$ is doubly stochastic, it can be interpreted as encoding the transitions of a Markov process. Spectral analysis produces the decomposition $\mathbf{K}\mathbf{V} = \mathbf{V}\mathbf{\Lambda}$, where $\mathbf{V} \in \mathbb{R}^{n \times n}$ denotes the eigenvector matrix with corresponding eigenvalues $\boldsymbol{\lambda} \in [0,1]^n$, and $\mathbf{\Lambda} = \text{diag}(\boldsymbol{\lambda})$. Indexing from zero, it can be shown that $V_0$ is constant, with $1 = \lambda_0 \geq \lambda_1 \geq \cdots \geq \lambda_n$. Following standard convention, we drop this uninformative dimension and take the leading eigenvectors from $V_1$.

The elements of this decomposition have several notable properties.[3] For instance, the resulting eigenvectors uniquely solve the constrained optimization problem:

$$\min_{\mathbf{V} \in \mathbb{R}^{n \times d_{\mathcal{Z}}}} \sum_{i,j} k_{ij} \|\mathbf{v}_i - \mathbf{v}_j\|_2 \quad \text{s.t. } \mathbf{V}^\top \mathbf{V} = \mathbf{I},$$

---

[2]Even in the unsupervised case, we typically train the ensemble with a regression or classification objective. The trick is to construct some $Y$ that encourages the model to make splits that are informative w.r.t. $P_X$ (e.g., [85, 99]). For fully random partitions, $Y$ can be any variable that is independent of the features (e.g., [41, 40]).

[3]Observe that the eigenvectors of $\mathbf{K}$ are identical to those of the graph Laplacian $\mathbf{L} = \mathbf{I} - \mathbf{K}$, which has $j$th eigenvalue $\gamma_j = 1 - \lambda_j$. For more on the links between diffusion and Laplacian eigenmaps, see [62, 52].

for all $d_{\mathcal{Z}} \in [n]$, thereby minimizing Dirichlet energy and producing the smoothest possible representation of the data that preserves local relationships in the graph. These eigenvectors also simplify a number of otherwise intractable graph partition problems, providing smooth approximations that motivate spectral clustering approaches [84, 97]. If we think of $\mathcal{G}_n$ as a random sample from a Riemannian manifold $\mathcal{M}$, then scaling each $V_j$ by $\sqrt{n}\lambda_j^t$ produces an approximation of the $j$th eigenfunction of the Laplace-Beltrami operator at time $t$, which describes how heat (or other quantities) diffuse across $\mathcal{M}$ [8, 10]. Euclidean distance in the resulting space matches diffusion distances across $\mathcal{G}_n$, providing a probabilistically meaningful embedding geometry [26].

The diffusion map $\mathbf{Z} = \sqrt{n}\mathbf{V}\mathbf{\Lambda}^t$ represents the long-run connectivity structure of the graph after $t$ time steps of a Markov process. Test data can be projected into spectral space via the Nyström formula [51], i.e. $\mathbf{Z}_0 = \mathbf{K}_0\mathbf{Z}\mathbf{\Lambda}^{-1}$ for some $\mathbf{K}_0 \in [0,1]^{m \times n}$, where rows index test points and columns index training points. For more details on diffusion and spectral graph theory, see [24, 74, 97].

## 4 Decoding

Our next task is to solve the inverse problem of decoding vectors from the spectral embedding space back into the original feature space—i.e., learning the function $h : \mathcal{Z} \mapsto \mathcal{X}$ such that $\mathbf{x} \approx h\big(g(\mathbf{x})\big)$. We propose several solutions, including methods based on constrained optimization, split relabeling, and $k$-nearest neighbors. To study the properties of these different methods, we introduce the notion of a universally consistent decoder.

**Definition 4.1** (Universally consistent decoder). Let $g^* : \mathcal{X} \mapsto \mathcal{Z}$ be a lossless encoder. Then we say that the sequence of decoders $\{h_n : \mathcal{Z} \mapsto \mathcal{X}\}$ is *universally consistent* if, for all distributions $P_X$, any $\mathbf{x} \sim P_X$, and all $\epsilon > 0$, we have:

$$\lim_{n \to \infty} P\Big( \|\mathbf{x} - h_n\big(g^*(\mathbf{x})\big)\|_\infty \geq \epsilon \Big) = 0.$$

The first two methods—constrained optimization and split relabeling—are designed to infer likely leaf assignments for latent vectors $\mathbf{z}$. If these are correctly determined, then the intersection of assigned leaves defines a bounding box that contains the corresponding input vector $\mathbf{x}$. Our estimate $\hat{\mathbf{x}}$ is then sampled uniformly from this subspace, which is generally small for sufficiently large and/or deep forests. As a motivation for this approach, we show that a leaf assignment oracle would constitute a universally consistent decoder.

Let $d_\Phi^{(b)}$ be the number of leaves in tree $b$, and $d_\Phi = \sum_{b=1}^B d_\Phi^{(b)}$ the number of leaves in the forest $f$. The function $\pi_f : \mathcal{X} \mapsto \{0,1\}^{d_\Phi}$ maps each sample $\mathbf{x}$ to its corresponding leaves in $f$. It is composed by concatenating the outputs of $B$ unique functions $\pi_f^{(b)} : \mathcal{X} \mapsto \{0,1\}^{d_\Phi^{(b)}}$, each satisfying $\|\pi_f^{(b)}(\mathbf{x})\|_1 = 1$ for all $\mathbf{x} \in \mathcal{X}$. Let $\psi_f : \mathcal{Z} \mapsto \{0,1\}^{d_\Phi}$ be a similar leaf assignment function, but for latent vectors. Then for a fixed forest $f$ and encoder $g$, the *leaf assignment oracle* $\psi_{f,g}^*$ satisfies $\pi_f(\mathbf{x}) = \psi_{f,g}^*\big(g(\mathbf{x})\big)$, for all $\mathbf{x} \in \mathcal{X}$.

**Theorem 4.2** (Oracle consistency). *Let $f_n$ be a RF trained on $\{\mathbf{x}_i, y_i\}_{i=1}^n \overset{i.i.d.}{\sim} P_{XY}$. Let $h_n^* : \mathcal{Z} \mapsto \mathcal{X}$ be a decoder that (i) maps latent vectors to leaves in $f_n$ via the oracle $\psi_{f_n,g}^*$; then (ii) reconstructs data by sampling uniformly from the intersection of assigned leaves for each sample. Then, under the assumptions of Thm. 3.4, the sequence $\{h_n^*\}$ is universally consistent.*

### 4.1 Constrained Optimization

Our basic strategy for this family of decoders is to estimate a kernel matrix from a set of embeddings, then use this matrix to infer leaf assignments. Let $\mathbf{s} \in \{1/[n-1]\}^{d_\Phi}$ be a vector of inverse leaf sample sizes, composed of tree-wise vectors $\mathbf{s}^{(b)}$ with entries $s_i^{(b)} = 1/\sum_{j=1}^n \pi_i^{(b)}(\mathbf{x}_j)$. Then the canonical feature map for the RF kernel can be written $\phi(\mathbf{x}) = \big[\phi^{(1)}(\mathbf{x}), \ldots, \phi^{(B)}(\mathbf{x})\big]$, with tree-wise feature maps $\phi^{(b)}(\mathbf{x}) = \pi^{(b)}(\mathbf{x}) \odot \sqrt{\mathbf{s}^{(b)}}$, where $\odot$ denotes the Hadamard (element-wise) product. Now RF kernel evaluations can be calculated via the scaled inner product $k_n^{RF}(\mathbf{x}, \mathbf{x}') = B^{-1}\langle \phi(\mathbf{x}), \phi(\mathbf{x}') \rangle$, which is equivalent to Eq. 1.

Say we have $n$ training samples used to fit the forest $f_n$, and $m$ latent vectors to decode from an embedding space of dimension $d_{\mathcal{Z}} < n$. We will refer to these samples as a test set, since they may

not correspond to any training samples. We are provided a matrix of embeddings $\mathbf{Z}_0 \in \mathbb{R}^{m \times d_\mathcal{Z}}$, from which we estimate the corresponding kernel matrix $\hat{\mathbf{K}}_0 = \mathbf{Z}_0 \mathbf{\Lambda} \mathbf{Z}^\dagger$, where $\mathbf{Z}^\dagger$ denotes the Moore-Penrose pseudo-inverse of $\mathbf{Z}$.

Now we must identify the most likely leaf assignments for each of our $m$ (unseen) test samples $\mathbf{X}_0 \in \mathbb{R}^{m \times d_\mathcal{X}}$, given their latent representation $\mathbf{Z}_0$. Call this target matrix $\mathbf{\Psi} \in \{0, 1\}^{m \times d_\Phi}$. To estimate it, we start with the binary matrix of leaf assignments for training samples $\mathbf{\Pi} \in \{0, 1\}^{n \times d_\Phi}$. Exploiting the inner product definition of a kernel, observe that our original (training) adjacency matrix $\mathbf{K}$ satisfies $B\mathbf{K} = \mathbf{\Phi}\mathbf{\Phi}^\top = \mathbf{\Pi}\mathbf{S}\mathbf{\Pi}^\top$, where $\mathbf{\Phi} \in [0, 1]^{n \times d_\Phi}$ is the RKHS representation of $\mathbf{X}$, and $\mathbf{S} = \mathrm{diag}(\mathbf{s})$. We partition $[d_\Phi]$ into $B$ subsets $L^{(b)}$ that index the leaves belonging to tree $b$. Recall that each tree $b$ partitions the feature space $\mathcal{X}$ into $L^{(b)}$ hyperrectangular subregions $\mathcal{X}_\ell^{(b)} \subset \mathcal{X}$, one for each leaf $\ell \in [L^{(b)}]$. Let $R_i^{(b)} \in \{\mathcal{X}_1^{(b)}, \ldots, \mathcal{X}_{L^{(b)}}^{(b)}\}$ denote the region to which sample $i$ is routed in tree $b$. Then leaf assignments for test samples can be calculated by solving the following integer linear program (ILP):

$$\min_{\mathbf{\Psi} \in \{0,1\}^{m \times d_\Phi}} \|B\hat{\mathbf{K}}_0^\top - \mathbf{\Pi}\mathbf{S}\mathbf{\Psi}^\top\|_1 \quad \text{s.t. } \forall i, b : \sum_{\ell \in L^{(b)}} \psi_{i\ell} = 1, \quad \forall i : \bigcap_{b \in [B]} R_i^{(b)} \neq \emptyset. \quad (3)$$

The objective is an entry-wise $L_1$ norm that effectively treats the resulting matrix as a stacked vector. The first constraint guarantees that test samples are one-hot encoded on a per-tree basis. The second constraint states that the intersection of all assigned hyperrectangles is nonempty. We call these the *one-hot* and *overlap* constraints, respectively. Together they ensure consistent leaf assignments both within and between trees. The ILP approach comes with the following guarantee.

**Theorem 4.3** (Uniqueness). *Assume we have a lossless encoder $g^* : \mathcal{X} \mapsto \mathcal{Z}$ such that the estimated $\hat{\mathbf{K}}_0$ coincides with the ground truth $\mathbf{K}_0^*$. Then, under the assumptions of Thm. 3.4, as $n \to \infty$, with high probability, the ILP of Eq. 3 is uniquely solved by the true leaf assignments $\mathbf{\Psi}^*$.*

Together with Thm. 4.2, Thm. 4.3 implies that the ILP approach will converge on an exact reconstruction of the data under ideal conditions. While this may be encouraging, there are two major obstacles in practice. First, this solution scales poorly with $m$ and $d_\Phi$. Despite the prevalence of highly optimized solvers, ILPs are NP-complete and therefore infeasible for large problems. Second, even with an oracle for solving this program, results may be misleading when using noisy estimates of $\hat{\mathbf{K}}_0$. Since this will almost always be the case for $d_\mathcal{Z} \ll n$—an inequality that is almost certain to hold in most real-world settings—the guarantees of Thm. 4.3 will rarely apply. We describe a convex relaxation in Appx. C via the exclusive lasso [108, 19], an approach that is more tractable than the ILP but still turns each decoding instance into a nontrivial optimization problem.

## 4.2 Split Relabeling

Another strategy for computing leaf assignments front-loads the computational burden so that downstream execution requires just a single pass through a pretrained model, as with neural autoencoders. We do this by exploiting the tree structure itself, relabeling the splits in the forest so that they apply directly in embedding space.

The procedure works as follows. Recall that each split is a literal of the form $X_j \bowtie x$ for some $j \in [d_\mathcal{X}], x \in \mathcal{X}_j$, where $\bowtie \in \{<, =\}$ (the former for continuous, the latter for categorical data). At each node, we create a synthetic dataset $\tilde{\mathbf{X}}$ by drawing uniformly from the corresponding region. We embed these points with a diffusion map to create the corresponding matrix $\tilde{\mathbf{Z}}$. Samples are labeled with a binary outcome variable $Y$ indicating whether they are sent to left or right child nodes. Next, we search for the axis-aligned split in the embedding space that best predicts $Y$. The ideal solution satisfies $Z_k < z \Leftrightarrow X_j \bowtie x$, for some $k \in [d_\mathcal{Z}], z \in \mathcal{Z}_k$. Perfect splits may not be possible, but optimal solutions can be estimated with CART. Once all splits have been relabeled, the result is a tree with the exact same structure as the original but a new semantics, effectively mapping a recursive partition of the input space onto a recursive partition of the embedding space.

This strategy may fare poorly if no axis-aligned split in $\mathcal{Z}$ approximates the target split in $\mathcal{X}$. In principle, we could use any binary decision procedure to route samples at internal nodes. For example, using logistic regression, we could create a more complex partition of $\mathcal{Z}$ into convex polytopes. Of course, this increase in expressive flexibility comes at a cost in space and time complexity. The use

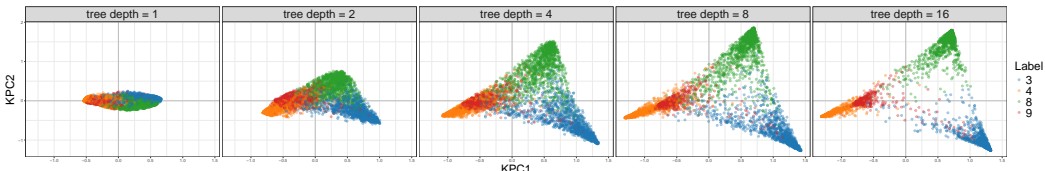

Figure 2: Diffusion maps visualize RF training. Using a subsample of the MNIST dataset, we find that digits become more distinct in the embedding space as tree depth increases.

of synthetic data allows us to choose the effective sample size for learning splits, which is especially advantageous in deep trees, where few training points are available as depth increases.

### 4.3 Nearest Neighbors

Our final decoding strategy elides the leaf assignment step altogether in favor of directly estimating feature values via $k$-nearest neighbors ($k$-NN). First, we find the most proximal points in the latent space. Once nearest neighbors have been identified, we reconstruct their associated inputs using the leaf assignment matrix $\mathbf{\Pi}$ and the splits stored in our forest $f_n$. From these ingredients, we infer the intersection of all leaf regions for each training sample—what Feng and Zhou [34] call the "maximum compatible rule"—and generate a synthetic training set $\tilde{\mathbf{X}}$ by sampling uniformly in these subregions. Observe that this procedure guarantees $g(\tilde{\mathbf{X}}) = \mathbf{Z}$ by construction.

Neighbors are weighted in inverse proportion to their diffusion distance from the target $\mathbf{z}_0$, producing the weight function $w : \mathcal{Z} \mapsto \Delta^{k-1}$. Let $K \subset [n]$ denote the indices of the selected points and $\{\tilde{\mathbf{x}}_i\}_{i \in K}$ the corresponding synthetic inputs. Then the $j$th entry of the decoded vector $\hat{\mathbf{x}}_0$ is given by $\hat{x}_{0j} = \sum_{i \in K} w_i(\mathbf{z}_0) \, \tilde{x}_{ij}$. For categorical features, we take the most likely label across all neighbors, weighted by $w$, with ties broken randomly. The $k$-NN decoder comes with similar asymptotic guarantees to the previous methods, without assuming access to a leaf assignment oracle.

**Theorem 4.4** ($k$-NN consistency). *Let $k \to \infty$ and $k/n \to 0$. Then under the assumptions of Thm. 3.4, the $k$-NN decoder is universally consistent.*

This approach is arguably truer to the spirit of RFs, using local averaging to decode inputs instead of just predict outputs. Like the split relabeling approach, this is a modular decoding solution that does not rely on any particular encoding scheme. However, it is uniquely well suited to spectral embedding techniques, which optimally preserve kernel structure with respect to $L_2$ norms in the latent space.

## 5 Experiments

In this section, we present a range of experimental results on visualization, reconstruction and denoising. Further details on all experiments can be found in Appx. B, along with additional results. Code for reproducing all results is available online.[4] The method is implemented as a package in R (see previous footnote), as well as in Python[5].

**Visualization** As a preliminary proof of concept, we visualize the embeddings of a RF classifier with 200 trees as it trains on a subset of the MNIST dataset [30] including samples with labels 3, 4, 8, and 9 (see Fig. 2). Plotting a sequence of diffusion maps with $d_{\mathcal{Z}} = 2$ at increasing tree depth, we find the model learning to distinguish between the four digits, which gradually drift into their own regions of the latent space. Early in training, the data are clumped around the origin. As depth increases, the manifold blooms and samples concentrate by class label. KPC1 appears to separate 3 and 8 from 4 and 9, which makes sense given that these respective pairs are often hard to distinguish in some handwritten examples. Meanwhile, KPC2 further subdivides 3 from 8. The relative proximity of 4's and 9's demonstrates that the RF is somewhat uncertain about these samples, although with extra dimensions we find clearer separation (not shown). In other words, the embeddings suggest a highly interpretable recursive partition, as we might expect from a single decision tree.

---

[4] https://github.com/bips-hb/RFAE.

[5] https://github.com/binhducvu/RFAE_py.

Table 1: Mean distortion (with standard errors) for each method and dataset, across all $d_{\mathcal{Z}}$ values used and all runs. Best result per row is bolded.

| Dataset | RFAE | TVAE | TTVAE | AE | VAE |
|---|---|---|---|---|---|
| abalone | **0.167 (0.002)** | 0.309 (0.005) | 0.260 (0.003) | 0.230 (0.025) | 0.211 (0.006) |
| adult | 0.326 (0.007) | **0.158 (0.005)** | 0.195 (0.007) | 0.401 (0.003) | 0.391 (0.004) |
| banknote | **0.100 (0.012)** | 0.312 (0.013) | 0.276 (0.023) | 0.724 (0.023) | 0.771 (0.013) |
| bc | 0.333 (0.003) | 0.564 (0.003) | 0.359 (0.005) | **0.287 (0.008)** | 0.578 (0.003) |
| car | 0.320 (0.011) | 0.195 (0.014) | **0.107 (0.015)** | 0.349 (0.012) | 0.313 (0.011) |
| churn | **0.352 (0.012)** | 0.603 (0.011) | 0.422 (0.014) | 0.861 (0.005) | 0.731 (0.006) |
| credit | **0.315 (0.004)** | 0.450 (0.005) | 0.375 (0.011) | 0.450 (0.005) | 0.456 (0.004) |
| diabetes | **0.479 (0.016)** | 0.726 (0.007) | 0.643 (0.014) | 0.799 (0.011) | 0.895 (0.004) |
| dry_bean | 0.137 (0.002) | 0.273 (0.002) | 0.303 (0.008) | **0.083 (0.014)** | 0.206 (0.001) |
| forestfires | **0.575 (0.008)** | 0.804 (0.003) | 0.705 (0.008) | 0.782 (0.007) | 0.790 (0.003) |
| hd | **0.432 (0.008)** | 0.582 (0.003) | 0.605 (0.006) | 0.892 (0.003) | 0.916 (0.002) |
| king | **0.308 (0.008)** | 0.352 (0.006) | 0.348 (0.008) | 0.377 (0.011) | 0.518 (0.004) |
| marketing | 0.292 (0.009) | 0.304 (0.005) | **0.259 (0.011)** | 0.357 (0.007) | 0.372 (0.004) |
| mushroom | 0.083 (0.001) | 0.093 (0.003) | **0.011 (0.003)** | 0.055 (0.004) | 0.035 (0.004) |
| obesity | **0.227 (0.008)** | 0.354 (0.004) | 0.299 (0.008) | 0.306 (0.009) | 0.358 (0.003) |
| plpn | **0.176 (0.006)** | 0.282 (0.006) | 0.224 (0.011) | 0.384 (0.013) | 0.410 (0.009) |
| spambase | 0.558 (0.005) | 0.825 (0.002) | 0.807 (0.003) | **0.446 (0.010)** | 0.784 (0.001) |
| student | **0.371 (0.002)** | 0.424 (0.001) | 0.426 (0.004) | 0.536 (0.003) | 0.551 (0.002) |
| telco | 0.177 (0.003) | 0.155 (0.003) | **0.091 (0.007)** | 0.128 (0.005) | 0.130 (0.005) |
| wq | **0.240 (0.005)** | 0.691 (0.008) | 0.759 (0.006) | 0.467 (0.019) | 0.708 (0.004) |
| Average Rank | **1.80** | 3.38 | 2.45 | 3.27 | 4.10 |

**Reconstruction** We limit our decoding experiments in this section to the $k$-NN method, which proved the fastest and most accurate in our experiments (for a comparison, see Appx. B.2). Henceforth, this is what we refer to as the RF autoencoder (RFAE).

As an initial inspection of RFAE's reconstruction behavior, we autoencode the first occurrence of each digit in the MNIST test set for varying latent dimensionalities $d_{\mathcal{Z}} \in \{2, 4, 8, 16, 32\}$ in Fig. 3. For this experiment, we fit an (unsupervised) completely random forest [15] with $B = 1000$ trees, train the encoder on full training data, project the test samples into $\mathcal{Z}$ via Nyström, and decode them back using $k = 50$ nearest neighbors. Although RFAEs are not optimized for image data, the reconstructions produce mostly recognizable digits even with very few latent dimensions, with outputs that partially correspond to the wrong class. Best results can be observed at $d_{\mathcal{Z}} = 32$, where the reconstructions appear quite similar to the originals. Additional results examining the influence of other parameters are presented in Appx. B.2.

Next, we compare RFAE's compression-distortion trade-off against two state-of-the-art neural architectures for autoencoding tabular data (TVAE and TTVAE), along with standard and variational autoencoders (AE and VAE, respectively) that are not optimized for this task. Although there are some other notable deep learning algorithms designed for tabular data (e.g., CTGAN [103], TabSyn [106], and TabPFN [56]), these do not come with inbuilt methods for decoding back to the input space at variable compression ratios. We also do not include eForest [34], another RF based autoencoder, because it only works with a fixed

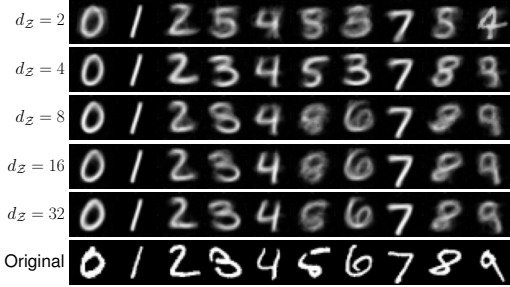

Figure 3: MNIST digit reconstructions with varying latent dimension sizes; original images are displayed in the bottom row.

$d_{\mathcal{Z}} = 2d_{\mathcal{X}}$ and is not capable of compression. For RFAE, we use the unsupervised ARF algorithm [99] with 500 trees and set $k = 20$ for decoding.

In standard AEs, reconstruction error is generally estimated via $L_2$ loss. This is not sensible with a mix of continuous and categorical data, so we create a combined measure that evaluates distortion on continuous variables via $1 - R^2$ (i.e., the proportion of variance unexplained) and categorical

variables via classification error. Since both measures are on the unit interval, so too is their average across all $d_{\mathcal{X}}$ features.

We measure this distortion across a range of compression factors (i.e., inverse compression ratios $d_{\mathcal{Z}}/d_{\mathcal{X}}$ in 20 benchmark tabular datasets. For more details on these datasets, see Appx. B.1, Table 2. We evaluate performance over ten bootstrap samples at each compression factor, testing on the randomly excluded out-of-bag data. We find that RFAE is competitive in all settings, and has best average performance in 12 out of 20 datasets (see Table 1), for an average rank of 1.80. For a more granular view of performance on each dataset, see Appx. B.2, Fig. 5.

**Denoising** As a final experiment, we consider a denoising example with single-cell RNA-sequencing (scRNA-seq) data. Pooling results from different labs is notoriously challenging in scRNA-seq due to technical artifacts collectively known as "batch effects" [65]. We propose to harmonize data across batches by training a RFAE on a large baseline study and passing new samples through the autoencoding pipeline.

As an illustration, we compare two studies of the mouse brain transcriptome. Using the top $d_{\mathcal{X}} = 5000$ genes, we learn a $d_{\mathcal{Z}} = 64$-dimensional embedding of the Zeisel et al. [105] dataset ($n = 2874$). Our RF is a completely random forest with $B = 1000$ trees. We project the Tasic et al. [91] data ($m = 1590$) into the latent space and decode using the top $k = 100$ nearest neighbors. Results are presented in Fig. 4. To avoid potential biases from reusing our own embeddings, we compare original and denoised samples using PCA [60] and tSNE [93],

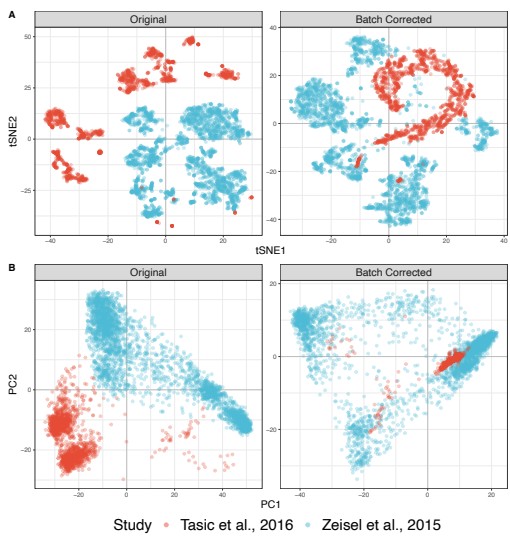

Figure 4: Denoising with RFAE alleviates batch effects in scRNA-seq data.

two dimensionality reduction techniques that are widely used in scRNA-seq. In both cases, we find that denoising with RFAE helps align the manifolds, thereby minimizing batch effects.

## 6 Discussion

The building blocks of our encoding scheme are well established. Breiman himself took a kernel perspective on RFs [16], a direction that has been picked up by numerous authors since [28, 83, 7]. The theory of diffusion maps and KPCA goes back some twenty years [25, 26, 80]. However, just as much RF theory has focused on idealized variants of the algorithm [14], no prior works appear to have studied the properties of the true RF kernel, opting instead to analyze simpler approximations. And while there have been some previous attempts to generate RF embeddings [85, 77], these have been largely heuristic in nature. By contrast, we provide a principled approach to dimensionality reduction in RFs, along with various novel decoding strategies.

One notable difference between our method and autoencoding neural networks is that RFAE is not trained end-to-end. That is, while a deep autoencoder simultaneously learns to encode and decode, RFAE is effectively a post-processing procedure for a pre-trained RF, with independent modules for encoding and decoding. We highlight that end-to-end training represents a fundamentally different objective. Whereas traditional autoencoders are necessarily unsupervised, our method can be readily applied to regression or classification forests, linking RFAE to supervised dimensionality reduction techniques [38, 6]. More generally, our goal is not just to learn an efficient representation for its own sake, but rather to reveal the inner workings of a target model. One upshot of this decoupling is that our split relabeling and $k$-NN decoders can work in tandem with any valid encoding scheme. For instance, we could relabel an RF's splits to approximate the behavior of sample points in principal component space, or indeed any $\mathcal{Z}$ for which we have a map $g : \mathcal{X} \mapsto \mathcal{Z}$.

We highlight two notable limitations of our approach. First, the computational demands of our decoding strategies are nontrivial. (For a detailed analysis, see Appx. D.) Second, when using the

$k$-NN approach, results will vary with the choice of $k$. (Experimental results on hyperparameter sensitivity are presented in Appx. B.2.) However, we observe that autoencoding is a difficult task in general. Top deep learning models generally pose far greater computational burdens than RFAE, and require many more hyperparameters to govern model architecture and regularization penalties. Compared to the leading alternatives, RFAEs are relatively lightweight and user-friendly.

Autoencoders are often motivated by appeals to the minimum description length principle [78, 55, 49]. Information theory provides a precise formalism for modeling the communication game that arises when one agent (say, Alice) wants to send a message to another (say, Bob) using a code that is maximally efficient with minimal information loss. This is another way to conceive of RFAE—as a sort of cryptographic protocol, in which Alice and Bob use a shared key (the RF itself) to encrypt and decrypt messages in the form of latent vectors $\mathbf{z}$, which is presumably more compact (and not much less informative) than the original message $\mathbf{x}$.

Several authors have persuasively argued that learning expressive, efficient representations is central to the success of deep neural networks [11, 44]. Our work highlights that RFs do something very similar under the hood, albeit through entirely different mechanisms. This insight has implications for how we use tree-based ensembles and opens up new lines of research for this function class.

# 7   Conclusion

We have introduced novel methods for encoding and decoding data with RFs. The procedure is theoretically sound and practically useful, with a wide range of applications including compression, clustering, data visualization, and denoising. Future work will investigate extensions to generative modeling, as well as other tree-based algorithms, such as gradient boosting machines [35, 22]. Another promising direction is to use the insights from this study to perform model distillation [54, 37], compressing the RF into a more compact form with similar or even identical behavior. This is not possible with current methods, which still require the original RF to compute adjacencies and look up leaf bounds.

## Acknowledgements

This research was supported by the UK Engineering and Physical Sciences Research Council (EPSRC) [Grant reference number EP/Y035216/1] Centre for Doctoral Training in Data-Driven Health (DRIVE-Health) at King's College London and by the German Research Foundation (DFG), Emmy Noether Grant 437611051.

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

# A  Proofs

Since several of our results rely on RF regularity conditions, we review these here for completeness.

We say that a sequence of functions $\{f_n\}$ is *universally consistent* if it converges in probability on any target function. That is, for any $f^* \in C(\mathcal{X})$ and all $\epsilon > 0$, we have:

$$\lim_{n \to \infty} P(\|f^* - f_n\|_\infty > \epsilon) = 0.$$

Under certain assumptions, it can be shown that RFs are universally consistent in this sense [70, 31, 14, 82, 98]. Specifically, we assume:

(A1) Training data for each tree is split into two subsets: one to learn split parameters, the other to assign leaf labels.

(A2) Trees are grown on subsamples rather than bootstraps, with subsample size $n^{(b)}$ satisfying $n^{(b)} \to \infty, n^{(b)}/n \to 0$ as $n \to \infty$.

(A3) At each internal node, the probability that a tree splits on any given $X_j$ is bounded from below by some $\rho > 0$.

(A4) Every split puts at least a fraction $\gamma \in (0, 0.5]$ of the available observations into each child node.

(A5) For each tree $b \in [B]$, the total number of leaves $d_\Phi^{(b)}$ satisfies $d_\Phi^{(b)} \to \infty, d_\Phi^{(b)}/n \to 0$ as $n \to \infty$.

Under (A1)-(A5), decision trees satisfy the criteria of Stone's theorem [89] and are therefore universally consistent (see Devroye et al. [32, Thm. 6.1] and Györfi et al. [50, Thm. 4.2]). The consistency of the ensemble follows from the consistency of the basis functions [15]. There is some debate in the literature as to whether these assumptions are necessary for universal consistency—(A1) and (A2) in particular may be overly strong—but they are provably sufficient. See Biau [13, Rmk. 8], Wager and Athey [98, Appx. B], and Tang et al. [90] for a discussion.

## A.1  Proof of Thm. 3.4 (RF kernel properties)

This theorem makes three separate claims: that RF kernels are (a) PSD and stochastic; (b) asymptotically universal; and (c) asymptotically characteristic.

**(a) PSD**  Take PSD first. It is well known that any convex combination of PSD kernels is PSD [79], so to secure part (a) it is sufficient to prove that the standard decision tree kernel is PSD. This is simply a normalized indicator kernel:

$$k^{DT}(\mathbf{x}, \mathbf{x}') = \frac{k^{(b)}(\mathbf{x}, \mathbf{x}')}{\sum_{i=1}^n k^{(b)}(\mathbf{x}, \mathbf{x}_i)},$$

which either evaluates to zero (if the samples do not colocate) or the reciprocal of the leaf sample size (if they do).

To show that $k^{DT}$ is PSD, we take a constructive approach in which we explicitly define the canonical feature map $\phi : \mathcal{X} \mapsto \mathcal{H}$, which maps input vectors to an inner product space $\mathcal{H}$. This suffices to establish the PSD property, since for any finite dataset the resulting kernel matrix $\mathbf{K}^{DT} \in [0, 1]^{n \times n}$ is a Gram matrix with entries $k_{ij}^{DT} = \langle \phi(\mathbf{x}_i), \phi(\mathbf{x}_j) \rangle$. As described in Sect. 4, the DT feature map for tree $b$ is given by:

$$\phi^{(b)}(\mathbf{x}) = \pi^{(b)}(\mathbf{x}) \odot \sqrt{\mathbf{s}^{(b)}},$$

where $\pi^{(b)} : \mathcal{X} \mapsto \{0, 1\}^{d_\Phi^{(b)}}$ is a standard basis vector indicating which leaf $\mathbf{x}$ routes to in tree $b$, and $\mathbf{s} \in \{1/[n-1]\}^{d_\Phi^{(b)}}$ is a vector of corresponding inverse leaf sample sizes. Concatenating these maps over $B$ trees and taking the inner product for sample pairs, we get an explicit formula for the RF feature map, thereby establishing that $k^{RF}$ is PSD.

It may not be immediately obvious that Eq. 1 is equivalent to the scaled inner product $B^{-1}\langle \phi(\mathbf{x}), \phi(\mathbf{x}') \rangle$. For completeness, we derive the identity. Expanding the inner product, we

have:

$$\langle \phi(\mathbf{x}), \phi(\mathbf{x}') \rangle = \sum_{b=1}^{B} \langle \phi^{(b)}(\mathbf{x}), \phi^{(b)}(\mathbf{x}') \rangle$$

$$= \sum_{b=1}^{B} \left\langle \pi^{(b)}(\mathbf{x}) \odot \sqrt{\mathbf{s}^{(b)}}, \pi^{(b)}(\mathbf{x}') \odot \sqrt{\mathbf{s}^{(b)}} \right\rangle$$

$$= \sum_{b=1}^{B} \sum_{\ell=1}^{L_b} \pi_\ell^{(b)}(\mathbf{x}) \, \pi_\ell^{(b)}(\mathbf{x}') \, s_\ell^{(b)}.$$

If both $\mathbf{x}$ and $\mathbf{x}'$ fall in leaf $\ell$ of tree $b$, then $\pi_\ell^{(b)}(\mathbf{x}) \, \pi_\ell^{(b)}(\mathbf{x}') = 1$; otherwise, the product evaluates to 0. Let $s_{\text{leaf}}^{(b)}$ denote the inverse sample size of the leaf containing $\mathbf{x}$ in tree $b$. Then $\langle \phi^{(b)}(\mathbf{x}), \phi^{(b)}(\mathbf{x}') \rangle = s_{\text{leaf}}^{(b)}$ if $\mathbf{x}, \mathbf{x}'$ fall in the same leaf of tree $b$, or 0 otherwise. Note that we have:

$$s_{\text{leaf}}^{(b)} = \frac{1}{\sum_{i=1}^{n} k^{(b)}(\mathbf{x}, \mathbf{x}_i)},$$

since the number of training points in the same leaf as $\mathbf{x}$ is given by $\sum_{i=1}^{n} k^{(b)}(\mathbf{x}, \mathbf{x}_i)$. Substituting and dividing by $B$ gives:

$$\frac{1}{B} \langle \phi(\mathbf{x}), \phi(\mathbf{x}') \rangle = \frac{1}{B} \sum_{b=1}^{B} \frac{k^{(b)}(\mathbf{x}, \mathbf{x}')}{\sum_{i=1}^{n} k^{(b)}(\mathbf{x}, \mathbf{x}_i)},$$

which completes the derivation.

Part (a) makes an additional claim, however—that $k_n^{RF}$ is *stochastic*. This means that, for any $\mathbf{x} \in \mathcal{X}$, the kernel $k_n^{RF}(\mathbf{x}, \mathbf{x}_i)$ defines a probability mass function over the training data as $i$ ranges from 1 to $n$. It is easy to see that the kernel is nonnegative, as all entries are either zero (if samples do not colocate) or some positive fraction representing average inverse leaf sample size across the forest (if they do). All that remains then is to show that the values sum to unity. Consider the sum over all $n$ training points:

$$\sum_{i=1}^{n} k_n^{RF}(\mathbf{x}, \mathbf{x}_i) = \sum_{i=1}^{n} \frac{1}{B} \sum_{b=1}^{B} \left( \frac{k^{(b)}(\mathbf{x}, \mathbf{x}_i)}{\sum_{j=1}^{n} k^{(b)}(\mathbf{x}, \mathbf{x}_j)} \right)$$

$$= \frac{1}{B} \sum_{b=1}^{B} \sum_{i=1}^{n} \left( \frac{k^{(b)}(\mathbf{x}, \mathbf{x}_i)}{\sum_{j=1}^{n} k^{(b)}(\mathbf{x}, \mathbf{x}_j)} \right)$$

$$= \frac{1}{B} \sum_{b=1}^{B} \frac{\sum_{i=1}^{n} k^{(b)}(\mathbf{x}, \mathbf{x}_i)}{\sum_{j=1}^{n} k^{(b)}(\mathbf{x}, \mathbf{x}_j)}$$

$$= \frac{1}{B} \sum_{b=1}^{B} 1$$

$$= 1.$$

Since the kernel is symmetric, the training matrix $\mathbf{K} \in [0,1]^{n \times n}$ of kernel entries is doubly stochastic (i.e., all rows and columns sum to one).

**(b) Universal** Recall that the Moore-Aronszajn theorem tells us that every PSD kernel defines a unique RKHS [1]. Given that $k_n^{RF}$ is PSD, universality follows if we can show that the associated RKHS $\mathcal{H}$ is dense in $C(\mathcal{X})$. Of course, this is provably false at any fixed $n$, as $d_\Phi = o(n)$ by (A5), and a finite-dimensional $\mathcal{H}$ necessarily contains "gaps"—i.e., some functions $f^* \in C(\mathcal{X})$ such that $\langle f^*, h \rangle = 0$ for all $h \in \mathcal{H}$.

However, as $n$ and $d_\Phi$ grow, these gaps begin to vanish. Since these two parameters increase at different rates, and it is the latter that more directly controls function complexity, we interpret the subscript $\ell$ on $\mathcal{H}_\ell$ as indexing the leaf count. (As noted above, we focus on the single tree case, as ensemble consistency follows immediately from this.)

The following lemma sets up our asymptotic universality result.

**Lemma A.1** (RF subalgebra). *Let $\mathcal{H}_\ell$ denote the set of continuous functions on the compact metric space $\mathcal{X}$ representable by a tree with $\ell$ leaves, trained under regularity conditions (A1)-(A5). Define:*

$$\mathcal{A} := \bigcup_{\ell=1}^{\infty} \mathcal{H}_\ell.$$

*Then $\mathcal{A}$ is a subalgebra of $C(\mathcal{X})$ that contains the constant function and separates points.*

*Proof.* The lemma makes three claims, each of which we verify in turn.

(1) *$\mathcal{A}$ is a subalgebra of $C(\mathcal{X})$.*

Each $\mathcal{H}_\ell$ consists of continuous, piecewise constant functions on $\mathcal{X}$, induced by recursive binary partitions of $\mathcal{X}$ into $\ell$ leaf regions. These spaces are closed under addition, scalar multiplication, and multiplication, as the sum or product of two piecewise constant functions is piecewise constant over the common refinement of their partitions. Since $\mathcal{A}$ is the union of these $\mathcal{H}_\ell$, it is closed under addition, multiplication, and scalar multiplication.

(2) *$\mathcal{A}$ contains the constant functions.*

This follows immediately, as any tree (and hence any $\mathcal{H}_\ell$ can represent constant functions—for instance, a trivial tree with no splits assigns the same value to all points.

(3) *$\mathcal{A}$ separates points.*

By regularity conditions (A3) and (A4), every coordinate has a nonzero probability $\rho > 0$ of being selected for splitting at any node, and every split allocates at least a fraction $\gamma \in (0, 0.5]$ of the points to each child node. Meinshausen [70, Lemma 2] has shown that, under these conditions, the diameter of each leaf goes to zero in probability. This amount to an asymptotic injectivity guarantee. Since $\mathcal{X}$ is compact, for any pair of distinct points $\mathbf{x}, \mathbf{x}' \in \mathcal{X}$, there exists an index $\ell$ such that some $f_\ell \in \mathcal{H}_\ell$ assigns different values to $\mathbf{x}$ and $\mathbf{x}'$. In other words, some tree in $\mathcal{A}$ is guaranteed to route $\mathbf{x}$ and $\mathbf{x}'$ to different leaves.

$\square$

We now invoke the Stone-Weierstrass theorem to conclude the density of $\mathcal{A}$.

**Theorem A.2** (Stone–Weierstrass). *Let $\mathcal{X}$ be a compact Hausdorff space. If a subalgebra $\mathcal{A}$ of $C(\mathcal{X})$ contains the constant functions and separates points, then $\mathcal{A}$ is dense in $C(\mathcal{X})$ with respect to the uniform norm.*

Combining Lemma A.1 with this classical result, we conclude that the sequence $\{\mathcal{H}_\ell\}$ converges on a universal RKHS.

**(c) Characteristic**   Item (c) follows from (b) under our definition of universality. This was first shown by Gretton et al. [45, Thm. 3] in the non-asymptotic regime (although Sriperumbudur et al. [87] prove that the properties can come apart under slightly different notions of universality). We adapt the result simply by substituting a sufficiently close approximation in $\mathcal{H}$, where proximity is defined w.r.t. the supremum norm. Since the existence of such a close approximation is guaranteed by (b), the sequence $\{\mathcal{H}_\ell\}$ is asymptotically characteristic.

## A.2   Proof of Thm. 4.2 (Oracle consistency)

This result follows trivially from the universal consistency of the RF algorithm itself. Any partition of $\mathcal{X}$ that satisfies regularity conditions (A1)-(A5) converges on the true joint distribution $P_X$ as $n, d_\Phi \to \infty$ (see [68, Thm. 1] and [27, Thm. 2]). Resulting leaves have shrinking volume, effectively converging on individual points $\mathbf{x}$ with coverage proportional to the density $p(\mathbf{x})$. Because there are no gaps in these leaves (i.e., no subregions of zero density), a weighted mixture of uniform draws—with weights given by the leaf coverage—is asymptotically equivalent to sampling from $P_X$ itself. A leaf assignment oracle would therefore be guaranteed to map each latent vector $\mathbf{z} \in \mathcal{Z}$ to the corresponding input point $\mathbf{x} \in \mathcal{X}$, since leaf assignments effectively determine feature values in the large sample limit.

## A.3 Proof of Thm. 4.3 (Uniqueness)

It is immediately obvious that when $\hat{\mathbf{K}}_0 = \mathbf{K}_0^*$, the true leaf assignment matrix $\mathbf{\Psi}^*$ drives the ILP objective to zero and automatically satisfies the one-hot and overlap constraints. Our task, therefore, is to demonstrate that no other binary matrix $\hat{\mathbf{\Psi}} \neq \mathbf{\Psi}^*$ can do the same.

For simplicity, consider just a single test point ($m = 1$). Let $\mathcal{F} \subset \{0, 1\}^{d_\Phi}$ denote the feasible region of leaf assignment vectors, i.e. all and only those that satisfy the one-hot and overlap constraints. A sufficient condition for our desired uniqueness result is that no two feasible leaf assignments produce the same kernel values. More formally, if there exist no $\psi, \psi' \in \mathcal{F}$ such that $\mathbf{\Pi S}\psi^\top = \mathbf{\Pi S}\psi'^\top$, then the map $\psi \mapsto \mathbf{\Pi S}\psi^\top$ is injective over $\mathcal{F}$, in which case only a single solution can minimize the ILP objective.

Note that this injectivity property cannot be shown to hold in full generality. As a minimal counterexample, consider a case where the feature space is a pair of binary variables $\mathcal{X} = \{0, 1\}^2$ and our forest contains just two trees, the first placing a single split on $X_1$ and the second placing a single split on $X_2$. Now, say our training data comprises just two points, $\mathbf{x} = [0, 0]$ and $\mathbf{x}' = [1, 1]$. We observe the kernel entries $k(\mathbf{x}_0, \mathbf{x}) = k(\mathbf{x}_0, \mathbf{x}') = 1/2$. In this case, we know that $\mathbf{x}_0$ colocates with each training sample exactly once, and therefore that it shares exactly one coordinate with each of our two training points. However, we do not have enough information to determine where these colocations occur, since both $[0, 1]$ and $[1, 0]$ are plausible feature vectors for $\mathbf{x}_0$.

Such counterexamples become increasingly rare as the forest grows more complex. As previously noted, under (A3) and (A4), leaf diameter vanishes in probability [70, Lemma 2]. This is why random forests are asymptotically injective, a property we exploited in the proof for part (b) of Thm. 3.4. A corollary of Meinshausen's lemma is that for all distinct points $\mathbf{x}, \mathbf{x}' \in \mathcal{X}$:

$$\lim_{n \to \infty} P\big(f_n(\mathbf{x}) = f_n(\mathbf{x}')\big) = 0.$$

By modus tollens, it follows that if function outputs converge on the same value, then corresponding inputs must be identical. This is important, since with fixed training labels $Y$, model predictions are fully determined by kernel evaluations via Eq. 2. If two distinct vectors $\psi, \psi'$ produce identical values when left-multiplied by $\mathbf{\Pi S}$, then the corresponding inputs $\mathbf{x}, \mathbf{x}'$ cannot be separated by $f_n$. Therefore, with probability tending toward 1 as sample size grows, we conclude that the ILP of Eq. 3 is uniquely solved by the true leaf assignments.

## A.4 Proof of Thm. 4.4 ($k$-NN consistency)

To secure the result, we must show that

(a) $\tilde{\mathbf{x}} \xrightarrow{p} \mathbf{x}$: sampling uniformly from the intersection of each sample's assigned leaf regions asymptotically recovers the original inputs; and

(b) $\hat{\mathbf{x}} \xrightarrow{p} \tilde{\mathbf{x}}$: $k$-NN regression in the spectral embedding space is universally consistent.

Item (a), which amounts to a universal consistency guarantee for the eForest method of Feng and Zhou [34], follows immediately from our oracle consistency result (Thm. 4.2), plugging in the identity function for $g$. Since we know the true leaf assignments for each training sample, under universal consistency conditions for RFs, we can reconstruct the data by taking uniform draws from all leaf intersections.

Item (b) follows from the properties of the $k$-NN algorithm, which is known to be universally consistent as $k \to \infty, k/n \to 0$ [89].

## B  Experiments

RFs are trained either using the `ranger` package [101], or the `arf` package [99], which also returns a trained forest of class `ranger`. Truncated eigendecompositions are computed using `RSpectra`. Memory-efficient methods for sparse matrices are made possible with the `Matrix` package. We use the `RANN` package for fast $k$-NN regression with kd-trees. For standard lasso, we use the `glmnet` package [36], and `ExclusiveLasso` for the exclusive variant [19]. In both cases, we do not tune the

Table 2: Summary of datasets used. % Categorical indicates the proportion of categorical features. # Classes indicates the total cardinality of all the categorical features.

| Dataset | Code | #Samples | #Numerical | #Categorical | # Total | %Categorical | #Classes |
|---|---|---|---|---|---|---|---|
| Abalone | abalone | 4177 | 8 | 1 | 9 | 0.11 | 3 |
| Adult | adult | 45222 | 5 | 9 | 14 | 0.64 | 100 |
| Banknote Auth. | banknote | 1372 | 4 | 1 | 5 | 0.20 | 2 |
| Breast Cancer | bc | 570 | 30 | 1 | 32 | 0.03 | 2 |
| Car | car | 1728 | 0 | 7 | 7 | 1.00 | 25 |
| Bank Cust. Churn | churn | 10000 | 6 | 5 | 11 | 0.56 | 11 |
| German Credit | credit | 1000 | 7 | 14 | 21 | 0.67 | 56 |
| Diabetes | diabetes | 768 | 8 | 1 | 9 | 0.11 | 2 |
| Dry Bean | dry_bean | 13611 | 16 | 1 | 17 | 0.06 | 7 |
| Forest Fires | forestfires | 517 | 11 | 2 | 13 | 0.15 | 19 |
| Heart Disease | hd | 298 | 7 | 7 | 14 | 0.50 | 23 |
| King County Housing | king | 21613 | 19 | 0 | 19 | 0.00 | 0 |
| Bank Marketing | marketing | 45211 | 7 | 10 | 17 | 0.59 | 44 |
| Mushroom | mushroom | 8124 | 0 | 22 | 22 | 1.00 | 119 |
| Obesity Levels | obesity | 2111 | 8 | 8 | 16 | 0.50 | 30 |
| Palmer Penguins | plpn | 333 | 5 | 3 | 8 | 0.38 | 8 |
| Spambase | spambase | 4601 | 58 | 1 | 59 | 0.02 | 2 |
| Student Performance | student | 649 | 16 | 17 | 33 | 0.52 | 43 |
| Telco Churn | telco | 7032 | 3 | 17 | 20 | 0.85 | 43 |
| Wine Quality | wq | 4898 | 12 | 0 | 12 | 0.00 | 0 |

penalty parameter $\lambda$, but simply leave it fixed at a small value (0.0001 in our experiments). This is appropriate because we are not attempting to solve a prediction problem, but rather a ranking problem over coefficients.

## B.1 Reconstruction benchmark

We describe the setup for the experiments presented in the main text, Sect. 5.

For the compression reconstruction benchmark, we use 20 datasets sourced from the UCI Machine Learning Repository [33], OpenML [94], Kaggle, and the R palmerpenguins [57] package. In each dataset, we remove all rows with missing values. We also remove features that are not relevant to this task (e.g., customer names), features that duplicate in meaning (e.g., education and education_num in adult ) and date-time features. We report the subsequent in Table 2, describing for each used dataset the number of rows, features, and the proportion of categorical features in the total feature set, as a measure of the 'mixed'-ness of the tabular data.

Then, our pipeline proceeds as follows:

- For each dataset, we take ten different bootstrap samples, to form ten training sets, and use the remaining out-of-bag data as the testing set. We do this to ensure data is not unnecessarily unused, which could skew results in datasets with a small $n$.

- For each dataset, we define 10 compression ratios, or latent rates, going uniformly from 0.1 (10% of the number of features) to 1 (100% the number of features). For each value, we find $d_{\mathcal{Z}} = d_{\mathcal{X}} \times \text{latent\_rate}$

- For each dataset and bootstrap, we run each method with the specified $d_{\mathcal{Z}}$ for the dimensionality of their latent embeddings. This involves training the method on the bootstrap training data, then passing the testing data through its encoding and decoding stages to produce a reconstruction.

- We then compute the distortion of these reconstructions compared to the original test samples, using metrics described in the main text. For each bootstrap and $d_{\mathcal{Z}}$ on a dataset, we aggregate the results into a mean, and report the standard error as error bars. This standard error represents a combination of any stochastic component of the methods used, as well as the finite sample uncertainty of the used data.

Within this specification, we run each method $20 \times 10 \times 10 = 2000$ times, and for 5 methods this balloons to 10,000 runs. To meet these computational demands, we run these experiments from a high-performance computing partition, with 12 AMD EPYC 7282 CPUs, 64GB RAM, and an

NVIDIA A30 graphics card. These high-performance computing units were used as part of King's College London's CREATE HPC [67]

Next, we describe in detail each of the methods used in the reconstruction benchmark:

- **TVAE & TTVAE:** For both the TVAE and TTVAE models, we do not make any changes to the underlying model architecture. However, we make minor modifications to not perform any sampling/interpolation at the latent stage, and simply decode the same embeddings that we acquired from the encoder.

- **Autoencoder:** We use an MLP-based autoencoder for this benchmark. This is in contrast to a CNN-based autoencoder, which is more suited to image data, but is not relevant in tabular data tasks, where there is no inherent structure that could be extracted by convolution kernels. We structure our network to have five hidden layers, where the size of these layers are adaptive to $d_{\mathcal{X}}$ and $d_{\mathcal{Z}}$. In particular, we want to structure the network such that the size of each hidden layer reduces uniformly from $d_{\mathcal{X}}$ to $d_{\mathcal{Z}}$ at encoding and increases uniformly from $d_{\mathcal{Z}}$ to $d_{\mathcal{X}}$ at decoding. Our structure then is:

  Input($d_{\mathcal{X}}$) $\rightarrow$ Dense($d_{\mathcal{X}} - (d_{\mathcal{X}} - d_{\mathcal{Z}}) \times 1/3$) $\rightarrow$ Dense($d_{\mathcal{X}} - (d_{\mathcal{Z}} - d_{\mathcal{Z}}) \times 2/3$) $\rightarrow$ Latent($d_{\mathcal{Z}} d_{\mathcal{Z}}$) $\rightarrow$ Dense($d_{\mathcal{X}} - (d_{\mathcal{X}} - d_{\mathcal{Z}}) \times 2/3$) $\rightarrow$ Dense($d_{\mathcal{X}} - (d_{\mathcal{X}} - d_{\mathcal{Z}}) \times 1/3$) $\rightarrow$ Output($d_{\mathcal{X}}$).

  If $d_{\mathcal{X}} = 8, d_{\mathcal{Z}} = 2$ then with this rule the network will be $8 \rightarrow 6 \rightarrow 4 \rightarrow 2 \rightarrow 4 \rightarrow 6 \rightarrow 8$.

  For hyperparameters, we use common defaults: `epochs = 50`, `optimizer = ADAM`. We use ReLU activations at hidden layers, a sigmoid for the output, and a random 10% validation set from the training data.

- **Variational Autoencoder:** Similar to the autoencoder, we use an MLP-based variational autoencoder. For comparison, we mimic the autoencoder's architecture, activation function, and defaults, only changing `epochs = 100` and adding `batch_size = 32`. We also impose a $\beta$ coefficient inspired from [53], at a value of 0.1.

The RF training approach we use here is the adversarial random forests (ARFs) [99]. Using this algorithm allows us to train the forest unsupervised, and preserve all features. Other approaches involving unsupervised random forests can also be used, such as the ones proposed in Shi and Horvath [85] or Feng and Zhou [34]. However, we choose ARF on the basis that our decoders' complexity scale on $d_{\Phi}$, which is smaller for the ARF than others, because it learns its structure over several iterations of training.

We present the results in Fig. 5 corresponding to the plots in Table 1.

## B.2 Additional results

We present the results of additional experiments comparing different decoding methods, hyper-parameter analysis on MNIST, supervised vs. unsupervised RF embeddings as well as runtime experiments.

**Decoder Comparison** We compare the performance of three decoders that we describe in section 4: the kNN, the split relabelling and the LASSO decoder, on a smaller compression / reconstruction benchmark. We follow the same experimental setup as the previous experiment, but only use two datasets of small size (`credit` & `student`), and for each of these, we only use the first 5 bootstrap splits. This is because we only aim to illustrate the performance of these decoders side by side, and also because of the both the relabeling and LASSO's decoder much higher complexity compared to kNN. We maintain the **RFAE** setup as previously described in Appx. B.1. To make comparison easier, for each iteration we run the encoding stage once, and apply the three decoders to the same embeddings. We describe the relabeling and the LASSO decoder in more detail:

- **Relabeling Decoder:** For the relabeling decoder, we take the original RF object, and go through each split to find a corresponding split with the data on the latent space with the highest simple matching coefficient (SMC) to replace it. This process is global, so all data points are used in each split.

  An alternative approach to this is to re-find splits locally, i.e., only use the data located at each split in the original forest. However, in practice, the lack of training points at high

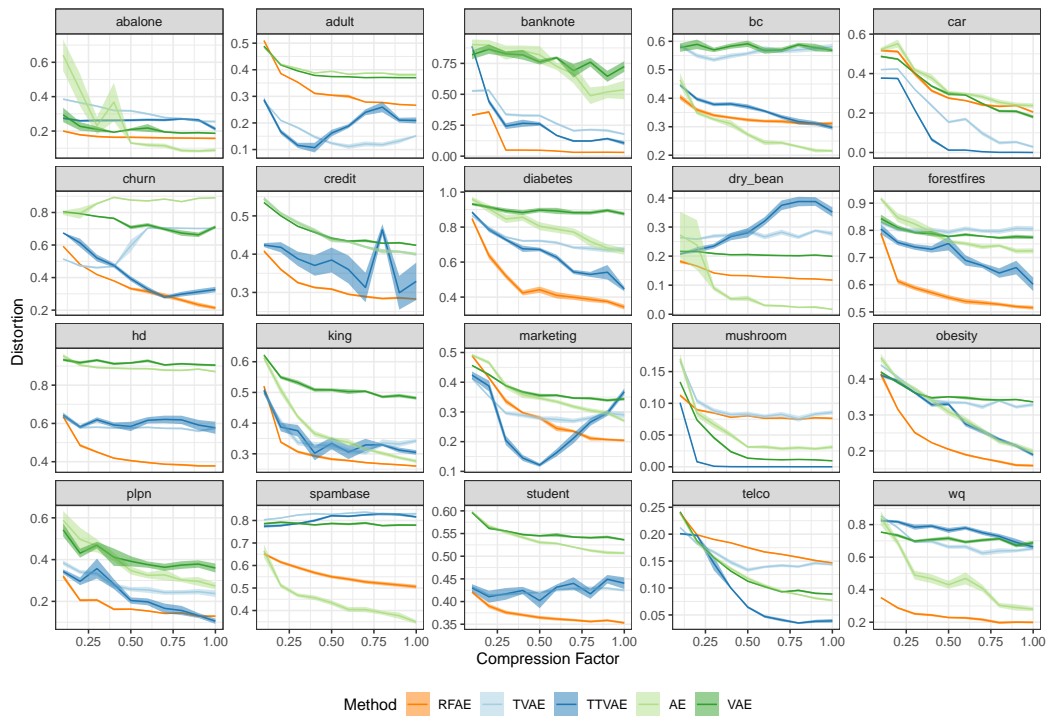

Figure 5: Compression-distortion trade-off on twenty benchmark tabular datasets. Shading represents standard errors across ten bootstraps.

depths will cause the relabeled splits to be inaccurate, so we use the global approach. Once the forest is 'relabeled', we can pass the embeddings for test data through it, and sample by using the leaf bounds of the original forest.

- **LASSO Decoder:** For the LASSO decoder, we follow the derivations in the main text and compute the kernel matrix $\hat{\mathbf{K}}_0 = \mathbf{Z}_0 \mathbf{\Lambda} \mathbf{Z}^\dagger$, and recover the leaf assignments for test samples via the LASSO & greedy leaf search algorithm in Appx. C. Once the leaf assignments are recovered, we once again sample by using the forest's leaf bounds.

  To reduce complexity, we impose a sparsity on the number of training samples allowed in the LASSO, sorted by the the values of the estimated kernel matrix. Because in most cases, a test sample will only land in the same leaf with a small number of training samples, even across an entire forest, this allows us to narrow the search space significantly with minimal impact on the model performance. For our datasets with 649 and 1000 samples, we set `sparsity = 100`.

We present the results in Fig. 6, where the kNN method is denoted as RFAE.

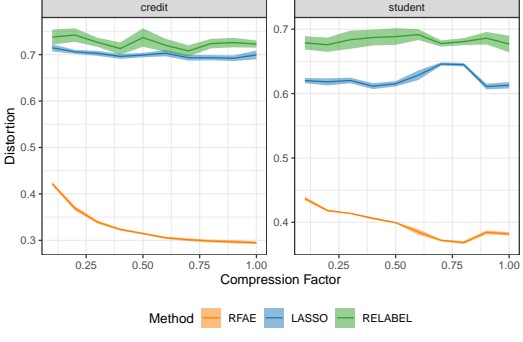

Figure 6: Compression-distortion trade-off on `student` and `credit`. Shading represents standard errors across five bootstraps.

From this plot, we can see that the kNN decoder dominates performance. Several reasons can explain the poor performance that the other decoders display. For the relabeling decoder, because our only criteria is to find the best matching split, if all candidates are bad, the best matching split does not have to be an objectively good choice. Given the forest's hierarchical nature, inaccuracies can be compounded traversing down the tree, and the final forest is completely dissimilar to the original. The small $d_{\mathcal{Z}}$ also means more variance is introduced into the process.

For the LASSO decoder, several things may have caused this performance. First, the estimation of $\hat{\mathbf{K}}$ may not be accurate, which is passed on to the LASSO optimization. The LASSO is also an approximate optimization, and variance here can lead to the greedy leaf algorithm to find the wrong leaf assignments for the test samples. This, in combination with the high complexity, motivate us to use the kNN decoder for **RFAE.**

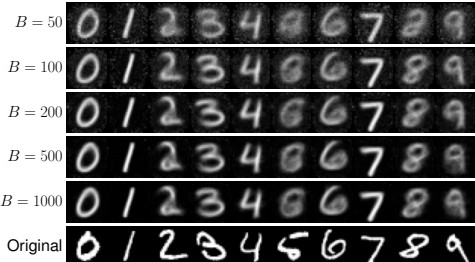

(a) Varying number of decision trees $B$ in random forest.

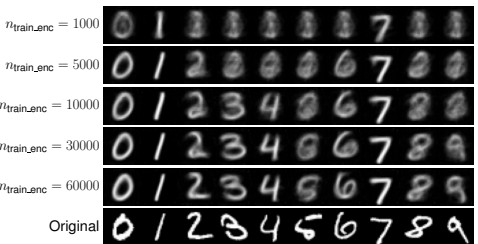

(b) Varying number of training points $n_{\text{train\_enc}}$ used for encoder training.

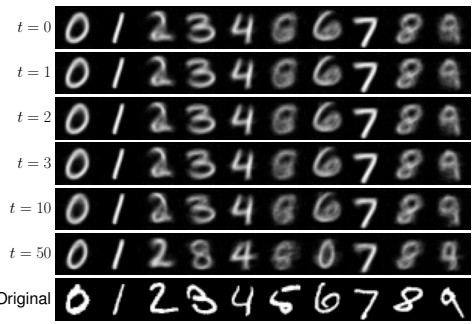

(c) Varying number of time steps $t$ for diffusion map encoding.

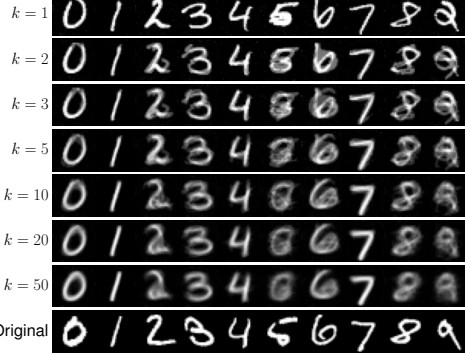

(d) Varying number of nearest neighbors $k$ used for $k$-NN-based decoding.

Figure 7: MNIST digit reconstructions produced by RFAE with varying parameter values; original images are displayed in the bottom row.

**MNIST reconstruction: hyperparameter analysis** Fig. 7 complements Fig. 3 in Section 5 by showing the effects of different parameters on the reconstruction performance for MNIST test digits: the number of trees $B$ (Fig. 7a), the number of samples $n_{\text{train\_enc}}$ used for encoder training (Fig. 7b), the diffusion time step $t$ (Fig. 7c), and the number $k$ of nearest neighbors used for decoding (Fig. 7d). In each subplot, the respective parameter varies in a pre-defined range, while all other ones are kept fixed at $B = 1000$, $n_{\text{train\_enc}} = 30\,000$, $t = 1$, and $k = 50$; the latent dimension is set to $d_{\mathcal{Z}} = 32$ throughout.

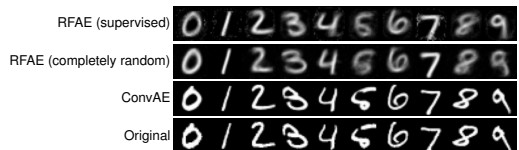

Figure 8: MNIST digit reconstructions produced by RFAE using supervised and copmletely random forests, and by a convolutional autoencoder; original images are displayed in the bottom row.

Fig. 8 compares RFAE reconstructions using supervised and (unsupervised) completely random forests with those produced by a convolutional autoencoder with three convolutional layers. All models were trained on full training data and with $d_{\mathcal{Z}} = 32$.

**Supervised vs. unsupervised embeddings**
RF embeddings provide different perspectives on the data depending on whether we use supervised or unsupervised learning objectives. We train a standard RF classifier to distinguish acute lymphoblastic (ALL) from acute myeloid (AML) leukemia samples in a well known transcriptomic dataset [43]. After filtering, the feature matrix includes expression values for $d_{\mathcal{X}} = 3572$ genes recorded in $n = 72$ patients. These kind of short, wide datasets are common in bioinformatics, but can be challenging for many learning algorithms. RFs excel in these

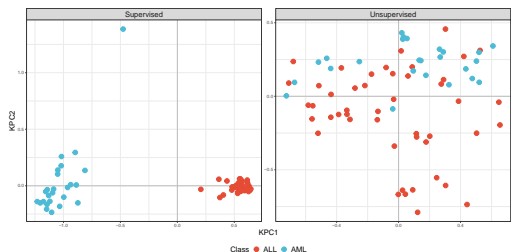

Figure 9: Embeddings can represent supervised or unsupervised RFs. Data from the Golub et al. [43] leukemia transcriptomics study.

settings, attaining out-of-bag accuracy of 98% on this task. KPC1 clearly separates the two classes in the left panel, while KPC2 appears to isolate a potential outlier within the AML cohort. Using an unsupervised adversarial RF [99], we find far greater overlap among the two classes (as expected), although ALL and AML samples are hardly uniform throughout the latent space. This example demonstrates the flexibility of our method. Whereas standard spectral embedding algorithms with fixed kernels are limited to unsupervised representations, RFAE can take any RF input.

**Runtime Experiments** Our runtime experiments consists of two tests; one where we compare the runtime of methods within the reconstruction benchmark (see B.1 for method and dataset details), and one where we exclusively examine the runtime of RFAE more closely. For the first experiment, we conduct a controlled runtime comparison of all five methods (AE, VAE, TVAE, TTVAE and RFAE) on three small-to-medium datasets (plpn, student, credit). These experiments are conducted on a laptop with Intel Core i5-10300H CPU, NVIDIA GeForce GTX 1650 (4GB), and 24GB DDR4 RAM. Each method was run 100 times across varying compression ratios, according to the same specifications as in the benchmark. We report the runtimes in Table 3, in seconds.

Table 3: Runtime comparison across five models and three datasets.

| Dataset | AE | VAE | RFAE | TVAE | TTVAE |
|---------|------|------|--------|--------|--------|
| plpn    | 161.6 | 185.9 | 523.8  | 744.2  | 2252.8 |
| student | 190.9 | 227.8 | 1058.6 | 3611.2 | 5148.7 |
| credit  | 223.2 | 277.3 | 1379.5 | 2212.5 | 4046.0 |

Here, the results indicate RFAE's superiority in runtime over the other two state-of-the-art methods in TVAE and TTVAE, while being slower than the naive AE and VAEs - which did poorly in the actual benchmark.

For the second experiment, we evaluate RFAE alone across 20 datasets, using a fixed compression ratio of 0.2 (10 runs per dataset). These were run on a HPC unit with an AMD Threadripper 3960X (24 cores, 48 threads) CPU and 256GB RAM, with no GPU unit used for the experiment. We report the **average** runtime for training and inference (as well as the total time), and their corresponding standard errors, ordered by average total runtime, in Table 4, in seconds. The results suggest RFAE scales well with dataset size.

## C Leaf assignments via lasso and greedy search

In this section, we briefly describe the lasso relaxation of the ILP in Eq. 3. First, we split the task into $m$ separate subproblems, one for each test vector. Let $\hat{\mathbf{k}}_0$ denote a row of $\hat{\mathbf{K}}_0$, say for test point $\mathbf{x}_0$. Observe that this adjacency vector is generally sparse, since most training points are unlikely to be neighbors of $\mathbf{x}_0$.[6] Let $a = \|\hat{\mathbf{k}}_0\|_0$ be the number of neighbors for $\mathbf{x}_0$, and write $\hat{\mathbf{k}}_0^{\downarrow} \in [0, 1]^a$ for the

---

[6]That neighbors vanish as a proportion of training samples is a common consistency condition for nonparametric models in general and local averaging estimators in particular [89].

Table 4: RFAE runtime performance across datasets (sorted by total mean).

| Dataset | Training $\mu$ | Training $\sigma$ | Inference $\mu$ | Inference $\sigma$ | Total $\mu$ | Total $\sigma$ | Samples | # Features |
|---|---|---|---|---|---|---|---|---|
| plpn | 1.89 | 0.00 | 1.35 | 0.01 | 3.24 | 0.03 | 333 | 8 |
| credit | 2.82 | 0.32 | 3.18 | 0.01 | 6.01 | 0.41 | 1,000 | 21 |
| car | 2.90 | 0.00 | 2.87 | 0.00 | 5.77 | 0.00 | 1,728 | 7 |
| student | 3.03 | 0.41 | 3.30 | 0.03 | 6.33 | 0.63 | 649 | 33 |
| diabetes | 3.16 | 0.01 | 3.97 | 0.01 | 7.13 | 0.01 | 768 | 9 |
| hd | 4.17 | 0.00 | 5.99 | 0.00 | 10.16 | 0.00 | 298 | 14 |
| forestfires | 4.63 | 0.30 | 6.12 | 0.02 | 10.76 | 0.36 | 517 | 13 |
| bc | 6.88 | 1.11 | 2.65 | 0.01 | 9.53 | 1.17 | 570 | 32 |
| abalone | 6.82 | 0.02 | 5.58 | 0.03 | 12.39 | 0.03 | 4,177 | 9 |
| wq | 7.90 | 0.73 | 6.38 | 0.01 | 14.28 | 0.81 | 4,898 | 12 |
| obesity | 8.74 | 0.02 | 12.11 | 0.08 | 20.85 | 0.05 | 2,111 | 16 |
| churn | 12.28 | 0.18 | 11.75 | 0.08 | 24.03 | 0.24 | 10,000 | 11 |
| mushroom | 13.80 | 0.23 | 11.53 | 0.10 | 25.33 | 0.48 | 8,124 | 22 |
| telco | 13.96 | 0.01 | 16.45 | 0.05 | 30.41 | 0.09 | 7,032 | 20 |
| spambase | 29.52 | 0.90 | 27.78 | 0.51 | 57.30 | 1.12 | 4,601 | 59 |
| dry_bean | 43.19 | 5.09 | 19.63 | 0.10 | 62.82 | 5.25 | 13,611 | 17 |
| king | 54.31 | 42.78 | 32.04 | 1.87 | 86.36 | 44.36 | 21,613 | 19 |
| adult | 150.27 | 304.30 | 89.47 | 24.85 | 239.75 | 469.52 | 45,222 | 14 |
| marketing | 148.96 | 4831.41 | 91.50 | 374.82 | 240.46 | 7889.97 | 45,211 | 17 |

reduction of $\hat{\mathbf{k}}_0$ to just its nonzero entries. We also write $L^{(b)\downarrow} \subseteq L^{(b)}$ for the set of leaves to which $\mathbf{x}_0$'s neighbors are routed in tree $b$, with cardinality $d_\Phi^{(b)\downarrow} \le d_\Phi^{(b)}$, and $d_\Phi^\downarrow = \sum_b d_\Phi^{(b)\downarrow}$. (Though the reduction operation is defined only w.r.t. some $\mathbf{x}_0$, we suppress the dependency to avoid clutter.) This implies corresponding reductions of $\mathbf{\Phi}$ to the submatrix $\mathbf{\Phi}^\downarrow \in \{0,1\}^{k \times d_\Phi^\downarrow}$, with one row per neighbor of $\mathbf{x}_0$ and columns for each leaf to which at least one neighbor is routed in $f$; and $\mathbf{S}$ to $\mathbf{S}^\downarrow \in [0,1]^{d_\Phi^\downarrow \times d_\Phi^\downarrow}$, with diagonal entries for each leaf in $\bigcup_b L^{(b)\downarrow}$.

Following these simplifications, we solve:

$$\min_{\boldsymbol{\psi} \in [0,1]^{d_\Phi^\downarrow}} \| B\hat{\mathbf{k}}_0^{\downarrow\top} - \mathbf{\Phi}^\downarrow \mathbf{S}^\downarrow \boldsymbol{\psi}^\top \|_2^2 + \lambda \sum_{b \in [B]} \Big( \sum_{\ell \in L^{(b)\downarrow}} \psi_\ell \Big)^2, \tag{4}$$

where the penalty factor $\lambda$ promotes a sparse solution with a similar effect to the one-hot constraint above. Specifically, it encourages competition both within trees (via the $L_1$ norm), and between trees (via the $L_2$ norm). Eq. 4 is an *exclusive lasso* problem, which can be efficiently solved via coordinate descent [108, 19] or dual Newton methods [71]. The interval (rather than integer) constraints on our decision variables effectively allow for "fuzzy" leaf membership, in which samples may receive nonzero weight in multiple leaves of the same tree.

Exploiting these fuzzy leaf assignments, we propose a greedy method to determine leaf membership. The method provisionally assigns a sample to the leaves with maximal entries in $\hat{\boldsymbol{\psi}}$. If any inconsistencies arise, we replace the "bad" assignments—i.e., those less likely to overlap with other assigned regions—with the next best leaves according to $\hat{\boldsymbol{\psi}}$. The procedure repeats until assignments are consistent. Though convergence is guaranteed, greedy search may prove inaccurate if coefficients are poorly estimated.

Alg. 1 provides an overview of the greedy leaf assignment procedure. This is run for a single sample $\mathbf{x} \in \mathcal{X}$, which has associated fuzzy leaf assignment vector $\hat{\mathbf{p}} \in [0,1]^{d_\Phi}$. We select the leaf coordinates associated with tree $b \in [B]$ by writing $\hat{\mathbf{p}}^{(b)} \in [0,1]^{d_\Phi^{(b)}}$. We overload notation somewhat by writing $R_i^{(b)} \subset \mathcal{X}$ to denote the hyperrectangular region associated with leaf $i \in [d_\Phi^{(b)}]$ of tree $b \in [B]$ in step 5; and then $R_q^{(b)}(t)$ to denote the region associated with leaf $q^{(b)}(t)$, which maximizes $\hat{\mathbf{p}}^{(b)}$ among all leaves that intersect with the feasible region $S(t)$. If the $\arg\max$ in step 5 is not unique, then we select among the maximizing arguments at random. Similarly, if there are multiple non-overlapping maximal cliques, then we select one at random in step 14. (This should be exceedingly rare in sufficiently large forests.) The undirected graph $\mathcal{G}(t)$ encodes whether the regions associated with assigned leaves overlap. Since consistent leaf assignments require intersecting regions for all trees, the algorithm terminates only when $\mathcal{G}(t)$ is complete. The procedure is greedy in the sense that edges

---

**Algorithm 1** GREEDYLEAFASSIGNMENTS

---

**Input**: Fuzzy leaf assignments $\hat{\mathbf{p}} \in [0, 1]^{d_\Phi}$

**Output**: Hard leaf assignments $\mathbf{q} \in \{1, \ldots, d_\Phi^{(b)}\}^B$

1: Initialize: $t \leftarrow 0$, $C(t) \leftarrow \emptyset$, $S(t) \leftarrow \mathcal{X}$, `converged` $\leftarrow$ `FALSE`
2: **while not** `converged` **do**
3:    $t \leftarrow t + 1$
4:    **for** all trees $b \in [B]$ **do**
5:       Find leaf with maximum fuzzy value in the feasible region:

$$q^{(b)}(t) \leftarrow \arg\max_{i \in [d_\Phi^{(b)}]} \hat{p}_i^{(b)} \quad \text{s.t.} \ \ R_i^{(b)} \cap S(t) \neq \emptyset$$

6:       Let $R_q^{(b)}(t)$ be the region corresponding to leaf $q^{(b)}(t)$
7:    **end for**
8:    Let $\mathcal{G}(t) = \langle [B], \mathcal{E}(t) \rangle$ be a graph with edges:

$$\mathcal{E}(t) := \{i, j \in [B] : R_q^{(i)}(t) \cap R_q^{(j)}(t) \neq \emptyset\}$$

9:    **if** $\mathcal{G}(t)$ is complete **then**
10:       `converged` $\leftarrow$ `TRUE`
11:    **else**
12:      **if** the maximal clique of $\mathcal{G}(t)$ is unique **then**
13:         Let $C(t) \subset [B]$ be the unique maximal clique of $\mathcal{G}(t)$
14:      **else if** the maximal cliques of $\mathcal{G}(t)$ have nonempty intersection **then**
15:         Let $C(t) \subset [B]$ be the intersection of all maximal cliques of $\mathcal{G}(t)$
16:      **else**
17:         Let $C(t) \subset [B]$ be any maximal clique of $\mathcal{G}(t)$
18:      **end if**
19:      Let $S(t) \leftarrow \bigcap_{b \in C(t)} R_q^{(b)}(t)$ be the feasible region associated with $C(t)$
20:    **end if**
21: **end while**
22: $\mathbf{q} \leftarrow \mathbf{q}(t)$

---

can only be added and never removed, leading to larger maximal cliques $C(t)$ and smaller feasible regions $S(t)$ with each passing round.

While the algorithm is guaranteed to converge on a consistent set of leaf assignments, this may only be achievable by random sampling of leaves under a consistency constraint. Such uninformative results are likely the product of noisy estimates for $\hat{\mathbf{K}}_0$. Note that while the maximal clique problem is famously NP-complete [61], we must solve this combinatorial task at most once (at $t = 1$). In subsequent rounds $t \geq 2$, we simply check whether any new nodes have been added to $C(t-1)$. Maximal clique solvers are highly optimized and have worst-case complexity $\mathcal{O}(2^{B/3})$, but are often far more efficient due to clever heuristics [102]. In practice, the method tends to find reasonable leaf assignments in just a few rounds. When computing leaf assignments for multiple test points, we may run Alg. 1 in parallel, as solutions are independent for each sample.

## D   Computational complexity

In this section, we study the complexity of different RFAE pipelines.

The encoding step requires $\mathcal{O}(n^2)$ space to store the adjacency matrix. For large datasets, this can be approximated by using a subset of the training data to compute $\mathbf{K}$, at the cost of losing the exact functional equivalence of Eq. 2. Similar tricks are common in the kernel methods literature [100, 107, 5].

While the ILP solution is generally intractable, the lasso relaxation requires $\mathcal{O}(d_\Phi^3)$ operations [19] to score leaves (although $d_\Phi$ can in fact be reduced to $a_i = \|\mathbf{k}_i\|_0$ for each test point $i \in [m]$; see Appx. C). The subsequent greedy leaf assignment algorithm searches for the maximal clique in a graph with $B$ nodes for each test sample, incurring worst-case complexity $\mathcal{O}\big(\exp(B)\big)$. Though the maximal clique problem is famously NP-complete [61], modern solvers often execute very quickly due to clever heuristics [102]. We can also parallelize over the test samples, as each greedy solution

is independent. Note that both the cubic term in the exclusive lasso task and the exponential term in the greedy leaf assignment algorithm are user-controlled parameters that can be reduced by training a forest with fewer and/or shallower trees. Alternatively, we could use just a subset of the RF's trees to reduce the time complexity of decoding via constrained optimization, much like we could use a subset of training samples to reduce the space complexity of encoding.

The split relabeling method is essentially an iterated CART algorithm, and therefore requires $\mathcal{O}(d_{\mathcal{Z}} \tilde{n} \log \tilde{n})$ time to relabel each split, where $\tilde{n}$ is the number of synthetic samples $\tilde{\mathbf{X}}$. However, since the number of splits is generally exponential in tree depth, this can still be challenging for deep forests.

The most efficient decoder is the $k$-NN method, which proceeds in two steps. First, we find the nearest neighbors in embedding space, which for reasonably small $d_{\mathcal{Z}}$ can be completed in $\mathcal{O}(m \log n)$ time using kd-trees. Next, we compute the intersection of all leaf regions for selected neighbors, incurring a cost of $\mathcal{O}(k m d_{\Phi} d_{\mathcal{X}})$.

