# OpenReview forum: "Autoencoding Random Forests"
_NeurIPS.cc/2025/Conference — NeurIPS 2025 poster_

### Official Review · Reviewer_B5CF · 2025-06-28

**Clarity:** 3
**Significance:** 3
**Originality:** 4
**Rating:** 5
**Confidence:** 2

**Summary:**

This paper proposes RFAE (Autoencoding Random Forests), a framework that turns a trained random forest into a data-dependent autoencoder.

The method constructs a kernel from the forest's structure, applies diffusion maps for encoding, and introduces three nonparametric decoders (ILP, k-NN, and split-relabel) to reconstruct the original input.

The paper also provides theoretical justification for the proposed kernel and decoder consistency.

Experiments show that RFAE achieves good reconstruction quality and offers interpretability benefits.

**Questions:**

1. How sensitive is the encoding-decoding quality to the number and depth of trees? How it depends on the RF quality?
2. Is there a way to extend this into a generative model? Any simple ideas?
3. Can the pipeline be simplified for users? For example, some kind of "default configuration" that works well for most datasets?

**Ethical Concerns:**

["NO or VERY MINOR ethics concerns only"]

**Final Justification:**

The responses address my earlier concerns on complexity, sensitivity to RF parameters, and potential generative extensions. In particular, the note that the k-NN decoder suffices in practice and that a default configuration works robustly across datasets improves my confidence in the method's practicality. I will maintain my accept recommendation.

**Limitations:**

yes

**Quality:**

3

**Strengths And Weaknesses:**

Pros:
- The paper tackles a pretty novel problem. It provides an explicit autoencoding framework based on random forests, and this encoder-decoder does not requires training a network. Training-free is especially interesting.
- Theoretical guarantees: The paper proves kernel universality and consistency/uniqueness (Theorems 3.4, 4.2-4), which strengthens its technical depth.
- Empirical results are convincing. The method shows competitive or better reconstruction quality compared to neural AEs/VAEs, especially on structured and small-scale tabular datasets.
- Interpretability: both embedding and reconstruction steps more transparent. You can reason about how the embedding and decoding work, which isn’t true for most neural approaches.

Cons:
- The pipeline is a bit complicated in practice. It involves RF training, spectral embedding (via diffusion maps), and then choosing among several decoder options. It increases implementation and deployment burden, and configs need to be tuned.
- The method is purely reconstructive. It can’t sample new data like VAEs.
- It’s quite dependent on the quality of the RF. This sensitivity is mentioned but not thoroughly analyzed.

---

> ### Author Rebuttal · Authors · 2025-07-30
>
> We sincerely thank reviewer B5CF (who we will refer to as R1) for their thoughtful and kind review. We address each of the comments below:
>
> -‘The pipeline is a bit complicated in practice…’: While we propose multiple decoder options, in our benchmarks, only the k-NN decoder is used for its simplicity and efficiency. In practice, this reduces the configuration to tuning the random forest (RF) and the number of neighbors for k-NN, making the method considerably easier to deploy than it may initially appear.
>
> -‘The method is purely reconstructive…’:  We concede that RFAE is not designed for data generation. This aligns with classical autoencoder frameworks (including neural methods), which focus on representation learning and reconstruction (as opposed to variational autoencoders, which the reviewer may have in mind). In principle, it could be possible to augment the model with a density estimator for the representation space, but this is outside the scope of our objectives.
>
> -‘It’s dependent of the quality of the RF’: We acknowledge that the reconstruction quality will generally depend on the quality of the underlying RF. However, we argue that any autoencoding method would inherently depend on the quality of its base model, and that in the mixed tabular data setting, RFs are generally a strong and efficient choice. Alternatively, we can view the RFAE as a way to characterize the quality and behaviour of the RF itself, as illustrated in Fig. 2.
>
> -‘How sensitive is the encoding-decoding quality…? How does it depend on the RF quality?’: Regarding sensitivity to depth, we did not constrain tree depth in our experiments and observed consistently strong performance. This suggests good model robustness and avoids introducing extra hyperparameters. For forest size, increasing the number of trees tends to improve performance up to a point. It would also increase the complexity of computing the kernel matrix $K$, increasing granularity and allowing RFAE to capture more complex relationships. Thus, the sensitivity to number of trees and RF quality should generally scale to the complexity of the task.
>
> -‘Is there a way to extend this into a generative model?’: We thank R1 for this interesting question. There exist generative RF models [1], but extending RFAE to synthetic data generation would require some method for sampling from the latent space. Since the latent space of RFAE is not parameterized (not multivariate normal, for instance, as in VAEs), this would presumably require generic density estimators such as KDE. Such ideas are doable in principle, but we reiterate from our above comment that it falls outside of our current objectives.
>
> -‘Can the pipeline be simplified…’ We appreciate this practical concern. A key strength of RFs is that they often perform well with minimal tuning. While we do modify settings across our different experiments (purely heuristically), within each experiment, we kept consistent hyperparameter values. Most notably, we used the same configuration for all datasets in our reconstruction benchmark (500 trees, the ranger default, and k=20), despite their highly variable structure and size. While it is likely the case that an optimal configuration exists for each dataset, our strong performance on this benchmark suggests that the settings we used can be a reasonable default configuration in implementation.
>
>
> References:
> [1]: https://proceedings.mlr.press/v206/watson23a.html

---

> > ### Author Response · Authors · 2025-08-05
> > **Deadline approaching**
> >
> > Thanks again to the reviewers for their time and consideration. Please let us know if there any further points in need of clarification. If our rebuttals have resolved all outstanding issues, then by all means feel free to revise scores upward :)

---

> > ### Comment · Reviewer_B5CF · 2025-08-05
> >
> > Thank you for the detailed rebuttal and clarifications. The responses address my earlier concerns on complexity, sensitivity to RF parameters, and potential generative extensions. In particular, the note that the k-NN decoder suffices in practice and that a default configuration works robustly across datasets improves my confidence in the method's practicality. I will maintain my accept recommendation.

---

> > > ### Author Response · Authors · 2025-08-06
> > > **Reply to B5CF**
> > >
> > > We are grateful to reviewer B5CF for their positive assessment. If our rebuttal "improves [the reviewer's] confidence in the method's practicality", then perhaps we could encourage B5CF to increase their confidence score? :)
> > >
> > > Thanks again for your time and consideration.

---

### Official Review · Reviewer_FErW · 2025-06-30

**Clarity:** 2
**Significance:** 3
**Originality:** 3
**Rating:** 4
**Confidence:** 2

**Summary:**

This paper proposes methods to embed data into compact representations with Random Forest kernels and diffusion maps. It also proposes methods to decode the embeddings back into the original input space, by utilizing methods like k-nearest neighbors. The resulting embedding can be used for downstream tasks.

**Questions:**

While the reconstruction results with MNIST benchmark appear interesting, they do not show how they compete with the similar results obtained by other baselines like deep neural networks.
The experiments reported in Figure 4 do not clearly show that the proposed method is a definitive winner. It would be helpful if the authors analyse when (under what benchmark characteristics) the proposed methods outperforms the other baselines. The authors should also supply computational time information.
In Figure 5, the authors claim that the denoising with RFAE helps align the manifolds, thereby minimizing batch effects. However, it is not clear by just looking at the figures. They should give more hints to the readers.

- I wonder if the defintion of the diffusion map should be something like $\boldsymbol{Z} = \sqrt{n}\boldsymbol{\Lambda}\boldsymbol{V}^\top$.

**Ethical Concerns:**

["NO or VERY MINOR ethics concerns only"]

**Final Justification:**

I have read the authors' response and other reviewers comments.
I acknowledge the novelty of the proposed decoder.
I will keep my score.

**Limitations:**

yes

**Paper Formatting Concerns:**

there are no major issues.

**Quality:**

3

**Strengths And Weaknesses:**

Strengths
+ The authors provide theoretical results including Theorem 3.4, which proves that the RF kernel is asymptotically universal and thus generally well-behaved.
+ The proposed RFAE methods can encode data which is a mix of continuous, ordinal and/or categorical variables.
+ The proposed RFAE is a post-processing procedure for a pre-trained RFs which are either supervised or unsupervised.
+ The authors have shown that RFs can encode and decode data effectively just like deep neural networks do but through entirely different mechanisms.

Weaknesses
- The proposed methods are computationally demanding.
- The experimental results do not clearly show the superiority of the proposed methods compared with the other baselines.

---

> ### Author Rebuttal · Authors · 2025-07-31
>
> We thank reviewer FErW (who we will refer to as R2) for their thoughtful review, and address all comments/questions below:
>
> -‘The proposed methods are computationally demanding’: We acknowledge that RFAE is not trivial in terms of computational cost. However, to the best of our knowledge, there are no existing methods for autoencoding mixed-type tabular data that are both faster and easier to implement while achieving comparable or better performance. State-of-the-art methods such as TTVAE [1] and TVAE [2] are also computationally intensive and, in our benchmarks (Figure 4 and Appx. B, Table 2), generally perform worse than RFAE in terms of reconstruction quality.
>
> -‘The experimental results do not clearly show the superiority…’: While Figure 4 provides a high-level overview, we agree that it may not make the overall ranking fully clear. RFAE outperforms all other methods on 12 out of 20 datasets and achieves the best average rank (1.8), with the next closest competitor, TTVAE, at 2.45 (Appx. B, Table 2). We will move this summary table into the main text in the final version to improve clarity. Although no single method dominates across all datasets, the results clearly favor RFAE.
>
> ‘While the reconstruction results with the MNIST benchmark appear interesting…’: We include comparisons with convolutional autoencoders in Appx. B.2 (Figure 8), and will move these to the main text in the final version. That being said, we note that random forests are not optimized for image data, and RFAE is not expected to compete with CNN-based autoencoders in this setting. Still, the performance remains respectable and supports the generality of our approach.
>
> ‘The experiments reported in figure 4 do not show that the proposed method is a definitive winner…’: We once again acknowledge that Figure 4 alone may have been difficult for readers to evaluate and refer R2 to Appx. B, Table 2 for a definitive summary.
>
> ‘The authors should also supply computational time information’: Thank you for this practical suggestion. We did not originally include runtime comparisons due to differences in implementation: RFAE is implemented in R and only runs on CPUs, while all other methods are implemented in Python and make use of GPU acceleration. Additionally, we ran experiments on high-performance computing infrastructure with varying configurations, making uniform comparison challenging.
>
>
> Regardless, in response to this comment, we have since conducted a controlled runtime comparison of all five methods (AE, VAE, TVAE, TTVAE, RFAE) on three small-to-medium datasets (plpn, student, credit), using the same hardware: a laptop with Intel Core i5-10300H CPU, NVIDIA GeForce GTX 1650 (4GB), and 24GB RAM. Each method was run 100 times across varying compression ratios, according to the same specifications as in the reconstruction benchmark. We report the results below, in seconds:
> | Dataset |   AE   |  VAE   |  RFAE  |  TVAE  | TTVAE  |
> |---------|--------|--------|--------|--------|--------|
> | plpn    | 161.6  | 185.9  | 523.8  | 744.2  | 2252.8 |
> | student | 190.9  | 227.8  | 1058.6 | 3611.2 | 5148.7 |
> | credit  | 223.2  | 277.3  | 1379.5 | 2212.5 | 4046.0 |
>
> Here, the results indicate RFAE’s superiority in runtime over the other two state-of-the-art methods in TVAE and TTVAE, while being slower than the naive AE and VAEs – which did very poorly in this benchmark.
>
> Additionally, we conducted a smaller-scale experiment evaluating RFAE alone across 18 datasets, using a fixed compression ratio of 0.2 (10 runs per dataset). Further datasets will be included in later camera-ready versions. We report the **average** runtime for training and inference (as well as the total time), and their corresponding standard errors, ordered by average total runtime, in seconds:
> | Dataset      | Training Mean | Training SE | Inference Mean | Inference SE | Total Mean | Total SE | #Samples | #Numerical | #Categorical | #Total |
> |--------------|----------------|-------------|------------------|----------------|-------------|-----------|-----------|-------------|----------------|--------|
> | plpn         | 3.714370       | 0.1332954   | 2.747021         | 0.2124611     | 6.461391   | 0.1932904 | 333       | 5           | 3              | 8      |
> | hd           | 3.822021       | 0.1437412   | 3.150167         | 0.2734484     | 6.972188   | 0.2667718 | 298       | 7           | 7              | 14     |
> | forestfires  | 4.450743       | 0.2971702   | 4.388958         | 0.1633418     | 8.839701   | 0.3532595 | 517       | 11          | 2              | 13     |
> | diabetes     | 4.732641       | 0.2556412   | 4.996549         | 0.2930842     | 9.729190   | 0.2435917 | 768       | 8           | 1              | 9      |
> | student      | 6.058696       | 0.4275493   | 6.718135         | 0.1241273     | 12.776831  | 0.4241161 | 649       | 16          | 17             | 33     |
> | credit       | 6.067794       | 0.2360919   | 7.017453         | 0.4312034     | 13.085247  | 0.6467874 | 1000      | 7           | 14             | 21     |
> | car          | 6.630258       | 0.2198116   | 6.909130         | 0.3269201     | 13.539389  | 0.4177712 | 1728      | 0           | 7              | 7      |
> | bc           | 12.171863      | 0.8757384   | 5.946692         | 0.2399783     | 18.118555  | 0.9651042 | 570       | 30          | 1              | 32     |
> | obesity      | 12.329533      | 0.3206506   | 9.920699         | 0.2508384     | 22.250232  | 0.5250324 | 2111      | 8           | 8              | 16     |
> | abalone      | 15.487057      | 0.1780608   | 11.296093        | 0.2203116     | 26.783150  | 0.3053351 | 4177      | 8           | 1              | 9      |
> | wq           | 17.698180      | 0.7578806   | 13.827139        | 0.3397825     | 31.525319  | 0.9755364 | 4898      | 12          | 0              | 12     |
> | mushroom     | 20.174338      | 0.3323456   | 24.847520        | 0.5722045     | 45.021858  | 0.8267018 | 8124      | 0           | 22             | 22     |
> | telco        | 27.032822      | 0.1789329   | 24.761390        | 0.5617190     | 51.794212  | 0.6939195 | 7032      | 3           | 17             | 20     |
> | churn        | 27.056492      | 0.3402298   | 30.225112        | 0.4099171     | 57.281603  | 0.5554646 | 10000     | 6           | 5              | 11     |
> | spambase     | 29.522823      | 0.9029132   | 27.781345        | 0.5143533     | 57.304168  | 1.1237604 | 4601      | 58          | 1              | 59     |
> | dry_bean     | 225.109069     | 8.0783953   | 45.221534        | 0.3427446     | 270.330602 | 7.9269351 | 13611     | 16          | 1              | 17     |
> | king         | 261.519581     | 13.9615926  | 103.128532       | 2.9978589     | 364.648112 | 14.8875831| 21613     | 19          | 0              | 19     |
>
> Here, the results suggests RFAE scales well with dataset size.
>
> -‘In Figure 5, the authors claim that the denoising with RFAE should help align the manifolds…’: To aid with interpretation, we have calculated the average distance between data points from each study before and after batch correction, in PCA space and tSNE space.
>
> |                | PCA     | tSNE    |
> |----------------|---------|---------|
> | Original       | 44.889  | 45.085  |
> | Batch-corrected| 22.289  | 35.661  |
>
> -‘I wonder if the definition of the diffusion map should be something like…’: This is very close to the precise definition that we offer at the end of Section 3 (line 147 of the manuscript), where we simply add the exponent $t$ to record the number of time steps in the Markov process. Since we keep this fixed at $t = 1$ in our experiments, our definition aligns with the one R2 proposes.
> References:
> [1]: https://www.sciencedirect.com/science/article/pii/S0004370225000116
> [2]: https://dl.acm.org/doi/10.5555/3454287.3454946

---

> > ### Comment · Reviewer_FErW · 2025-08-04
> >
> > Thank you for your response and explanation, and thanks for the additional experiments.
> > I will keep my score.

---

> > > ### Author Response · Authors · 2025-08-05
> > >
> > > Thanks for your comment. Please let us know if there any further points in need of clarification. If our rebuttals have resolved all outstanding issues, then by all means feel free to revise scores upward :)

---

> > ### Comment · Reviewer_FErW · 2025-08-06
> >
> > Thanks again for your response and explanation, and thanks for the additional experiments.
> > I acknowledge that the additional controlled runtime comparison shows the RFAE’s superiority in runtime over TVAE and TTVAE, which are also computationally demanding.
> > I also thank that the authors move the summary table into the main text in the final version to improve clarity.
> > It is encouraging that RFAE outperforms all other methods on 12 out of 20 datasets, while im some datasets, other methods work better. I would like to know if the authors can explain the reason why, in some particular cases, the competiter such as TTVAE is better.
> > I will keep my score, which is a positive reflection of the paper's overall quality and the authors' hard work.

---

> > > ### Author Response · Authors · 2025-08-06
> > > **Performance of TTVAE**
> > >
> > > Thank you for your new comments and questions.
> > >
> > > While investigating the exact behaviour of TTVAE is out of our scope, we do observe that it is a deep model with many parameters (e.g., 1028 hidden dimensions in their Transformer's feedforward network, and 300 epochs - see https://www.sciencedirect.com/science/article/pii/S0004370225000116 ). We noticed in experiments that this overparameterization gave it a tendency to overfit, making it worse than our method in most cases, particularly when numerical data is involved. However, since categorical datasets have a finite space of possible values, overfitting may appear less obvious in our evaluation metric (accuracy), leading to a better score.
> > >
> > > We are happy to answer any other questions that you may have. If these responses have resolved your current ones, we encourage you to consider revising your confidence score.

---

> > > > ### Comment · Reviewer_FErW · 2025-08-09
> > > >
> > > > Thanks for further explanations about the TTVAE results. If the inferior performance of TTVAE  is due to the excessed number of parameters, it may also mean that by adjusting the number of parameters (for example, reducing the number of hidden dimensions), the results may become different.
> > > >
> > > > As another note, the theoretical connection between the Spectral Graph Theory section and Section 4.1 was somewhat unclear and confusing to me, however, your response to another reviewer makes it clearer. I appreciate you revise the writing and make the reasoning clearer.

---

> ### Author Response · Authors · 2025-08-09
> **Modifying TTVAE parameters**
>
> Thank you for your response and further comments.
>
> During the benchmark, we do vary the dimensionality of the latent space, as we do with all models, to measure their reconstruction performance specifically. However, we kept the other settings at defaults. It is possible that altering the model parameters would reduce the overfitting issue, but we could make similar claims about every other model in the benchmark if we resort to manually tuning parameters. However, one of the purposes of the benchmark is to measure the robustness of each model, and their ability to perform under default configurations is an important aspect of this.
>
> Changing the network structure may also cause significant changes in behavior, affecting the validity of our results, so we avoided doing this in general. Finally, we should note that TTVAE's inferiority and overfitting comes within the context of comparison against RFAE - it is objectively not a poor model, as evidenced by its ranking within the benchmark.

---

### Official Review · Reviewer_58sa · 2025-07-03

**Clarity:** 2
**Significance:** 3
**Originality:** 2
**Rating:** 4
**Confidence:** 5

**Summary:**

The authors propose an autoencoder based on Kernel Principal Component Analysis, where a Random Forest is used to compute a kernel matrix.

Encoder: The kernel $k(x,y)$ is computed based on the proportion of trees in which samples $x$ and $y$ are routed to the same leaf, with a specific normalization applied. Then, the eigendecomposition of the resulting kernel matrix is computed, and the top eigenvectors are used to form the embedding. The proposed method does not have an exact out-of-sample mapping and relies on the Nystrom approximation.

Decoder: The algorithm does not have a natural way of decoding, so the authors propose several approximation methods:
- Constraint Optimization: This method uses an integer linear program to find the most likely leaf assignments for a given latent vector. After the optimal leaf in each tree is identified, the final reconstruction is generated by sampling a point uniformly from the intersection of all the corresponding leaf hyper-rectangles.
- Split Relabeling: This approach reconstructs the decision tree to operate in the low-dimensional latent space, such that instances follow the same route as in the original tree. Once a latent vector is routed to a final leaf in this new tree, the corresponding hyper-rectangle from the original tree is identified, and the reconstruction is sampled from that region.
- k-Nearest Neighbors (k-NN): This method, accelerated with kd-trees, finds the $k$ closest points in the latent space and identifies the corresponding regions in the original feature space. The final reconstruction is the weighted average of points sampled from these regions, with weights determined by the diffusion map.

The first two methods proved to be less accurate and more computationally intensive, so the experimental comparisons with other algorithms exclusively use the k-NN approach. Experiments are conducted on 20 tabular datasets with a maximum dimensionality of 59, single-cell RNA-sequencing data, and the MNIST dataset.

**Questions:**

- How is assumption (A3) enforced in practice, given that common RF implementations (e.g., scikit-learn, XGBoost) perform greedy split selection without guaranteed feature inclusion probabilities?
- If a split perfectly separates the classes but violates (A4), is it discarded? How are such constraints enforced in practical tree growth?
- Eq. (1) appears to normalize the kernel differently for each row, potentially making the matrix non-symmetric. Are the resulting eigenvalues guaranteed to be real and the matrix PSD?
- Can the decoder operate on new samples without needing access to the full training dataset? If not, how is this handled in deployment scenarios?
- Were the VAE models adequately tuned in terms of architecture and optimization? The result that a tree-based method outperforms VAEs on MNIST is surprising and needs stronger justification.

**Ethical Concerns:**

["NO or VERY MINOR ethics concerns only"]

**Final Justification:**

All of my concerns have been addressed. In particular:
- The authors clarified the necessity of a training dataset for out-of-sample mapping for the encoder.
- The authors provided a clearer explanation of the proof, which was unclear in the main paper.
- While the idea is not highly innovative, it presents an interesting approach to implementing autoencoders.

**Limitations:**

Discussed in weaknesses.

**Quality:**

2

**Strengths And Weaknesses:**

Strengths:
- The authors consider a fundamental problem of dimensionality reduction from the perspective of decision trees and provide a thorough theoretical justification for their work, proving several important properties of the adaptive RF kernel.

Weaknesses:

- The core idea is effectively a Kernel PCA where the kernel is derived from a Random Forest, which is not an entirely new concept. The use of trees as autoencoders is also not novel, as demonstrated by the recently proposed in [1].

- The proposed work does not have a true out-of-sample mapping capability and must rely on approximations. Furthermore, the most viable decoding method (k-NN) requires access to the entire original training dataset to reconstruct a new sample, which can be a significant limitation in practice.

- The authors' claim that "having just a single hyperparameter to worry about is a rare luxury in this field" is an oversimplification. The backbone of the proposed work is a Random Forest, which has many of its own hyperparameters that affect the final tree structures (e.g., depth, number of trees, splitting criterion). Additionally, the diffusion time step $t$ controls the scale of the embedding, and the k-NN decoder requires selecting $k$. The paper's own appendix shows that reconstructions vary significantly with the hyperparameters.

- A significant portion of the paper is dedicated to developing complex decoders based on constrained optimization (ILP and a LASSO relaxation), yet these methods are shown to be computationally inefficient and ultimately perform worse than the simple k-NN approach. This makes the most novel methodological contributions appear to have little practical value.

- The experiments are conducted on 20 tabular datasets where the maximum feature dimension is 59. This raises questions about the method's practicality and performance on truly high-dimensional tabular data, which is common in many real-world applications.

- The experimental evaluation lacks a crucial baseline comparison with Principal Component Analysis (PCA). While the authors focus on sophisticated neural network autoencoders, PCA serves as a fundamental, widely-used linear method for dimensionality reduction. Including PCA would have provided a performance floor and helped quantify the actual benefit of the proposed method's complexity, especially since calculating its reconstruction error is straightforward.

[1] M. Á. Carreira-Perpiñán and K. Gazizov. The tree autoencoder model, with application to hierarchical data visualization. (NeurIPS 25)

---

> ### Author Rebuttal · Authors · 2025-07-28
>
> We sincerely thank reviewer 58sa (henceforth R3) for their detailed review and thoughtful feedback. We reply to each comment/question below:
>
> -”The core idea is effectively a Kernel PCA...” We acknowledge that several authors have explored kernels and RFs before (see our copious bibliography) but want to emphasize that our primary contributions include (1) novel theoretical results establishing important properties of the RF kernel (e.g., that it is asymptotically universal and characteristic); and (2) various new methods for decoding these representations (we are unaware of any authors who have attempted this before).
>
> -“The use of trees as autoencoders is…not novel”: We thank R3 for bringing the tree autoencoder to our attention. We were unaware of this paper and will be sure to discuss it in the revised text. It appears that this method is only for continuous features, not mixed tabular data as we emphasize in our experiments. Moreover, the trees in question use half-plane splits as opposed to axis-aligned splits. This may be helpful in some settings but severs the connection to CART and RFs. One of our aims in this paper is to better understand this widely used class of supervised learners using the tools of nonparametric statistics and coding theory.
>
> -“Must rely on approximations”: We suspect that this comment conflates the Nyström method for approximating a Gram matrix (which we do not use) with the Nyström formula for embedding test points (which we do). The latter is  ubiquitous in autoencoding (see, e.g., IsoMAP, LLE, MDS, and more [1]) and is only “approximate” in the sense that the embedding space is low rank (but this is unavoidable in compression). With fixed $d_Z < n$, we are unaware of any more exact method for embedding test points. We are confused by R3’s judgment that RFAE “does not have a true out-of-sample mapping capability” – would R3 care to elaborate?
>
> -“requires access to the entire original training dataset”: We **do not** require the entire training dataset to reconstruct a new sample. All we need is the RF itself. From this, we create a synthetic dataset with the same leaf assignments as the training data, in the spirit of Feng & Zhou’s eForest [2]. This synthetic data is subsequently used for kNN, not the original training data. See the first paragraph of Sect. 4.3, where this is explicitly explained.
>
> -On hyperparameters: we thank R3 for highlighting this claim, which we agree amounts to an oversimplification. What we meant to say is that RFs are famously robust to hyperparameter selection and are known to work well with default settings in many different applications. Indeed, we did not tune any hyperparameters in our experiments. That said, we do use different numbers of trees for different tasks (heuristically, not systematically), and acknowledge some sensitivity to $k$ in our decoding. We have rewritten the discussion section to better reflect these points.
>
> -On constrained optimization: We begin our discussion of decoders with constrained optimization methods because of their attractive theoretical properties, which we suspect may tempt others to pursue this direction. However, as our work illustrates, there are more efficient alternatives. Proposing sound but intractable objectives before relaxing them into more feasible alternatives is standard practice in computer science. We are confused why R3 judges the kNN method to be of “little practical value” when it is universally consistent and far more efficient and accurate in practice than the ILP or split relabeling approach. We reiterate that we do not require the training data for kNN decoding but generate it on the fly using the forest structure. This novel methodological contribution goes on to fare well in a wide range of experiments.
>
> -On data dimensionality: We direct R3’s attention to our denoising experiment in the main text (Fig. 5), in which we use the top 5000 genes from two scRNA-seq datasets. In addition, we include a transcriptomics experiment with 3572 features in Appx. B2 (Fig. 9). Much like RFs themselves, our method scales well in high dimensions, especially when signals are relatively sparse.
>
> -On PCA: We thank R3 for inquiring about PCA, as our reason for excluding this baseline may have been unclear from the main text. The issue is that PCA is optimized for continuous data and struggles with categorical features, which must be one-hot encoded in preprocessing, artificially increasing dimensionality and making reconstruction a heuristic maximization procedure. As our main benchmark experiment focuses on mixed tabular data, we only considered methods that are specifically designed for handling such inputs. See Appx B, Table 2 for a summary of the breakdown between continuous/categorical features in these datasets. We can add PCA as a naive baseline, but would expect poor results as the proportion of categorical variables and number of category levels increase.
>
> -On assumption (A3): We acknowledge that (A3) may be violated when training a RF with greedy splits on some datasets, a point that should be addressed in the main text. However, three points are in order: (1) Such violations should be rare in the asymptotic regime ($n, B \rightarrow \infty$) provided that all features have nonzero entropy. Of course, this is still possible if a feature is fully independent of the outcome, but then it should arguably be excluded prior to model training. (2) In several experiments (including our MNIST experiment), we use completely random forests [2], which explicitly guarantee a uniform split probability on all features. (3) This assumption is ubiquitous in RF theory [3]. Though it may be a slight idealization, it is used to ground results for widely used implementations including of quantile forests [4] and generalized RFs [5]. That said, there is a persistent gap between RF theory and practice [6], a point we will address directly in the revised text.
>
> -On assumption (A4): We are unsure whether we understand R3’s question here. To violate (A4), a split must be “empty”, i.e. place 100% of the available data into one child node and 0% into the other. Such a split could not possibly “perfectly separate the classes”. Moreover, we are unaware of any standard RF implementation that permits such splits.
>
> -On PSD kernels: This is an important point, and we thank R3 for raising it. At first glance, Eq. 1 may appear asymmetric. However, the kernel is provably symmetric and PSD. For full details, see the proof in Appx. A1(a). The high-level intuition is that for any tree $b \in [B]$ and samples $\mathbf x, \mathbf x’ \in \mathcal X$, the kernel similarity $k^{(b)}(\mathbf x, \mathbf x’)$ must take one of two possible values: either (a) 0, if the samples do not colocate; or (b) the inverse leaf sample size, if they do. So if $\mathbf x, \mathbf x’$ colocate in a leaf to which 10 training points are routed, we would have $k^{(b)}(\mathbf x, \mathbf x’) = k^{(b)}(\mathbf x’, \mathbf x) = 1/10$. Again, we refer R3 to Appx. A1(a) for more details.
>
> -“Can the decoder operate on new samples without…access to the full training dataset?” Yes! See our comment above, and Sect. 4.3 for details. We generate a synthetic training set on the fly and use this for kNN, not the original training data itself.
>
> -“Were the VAE models adequately tuned…?” We are unsure which specific result R3 has in mind. We do not run any experiments with VAEs on MNIST. We do include a convolutional AE experiment on MNIST in the appendix (see Appx B.2, Fig. 8), where the neural decoder clearly outperforms RFAE, as expected. VAEs would presumably also do very well in this experiment. That said, we refer R3 to Feng & Zhou’s eForest paper [2], in which an RF based method for decoding performs very well on MNIST. (However, their method is not a proper autoencoder, for reasons we state in the main text.)
>
>
> References
>
> [1] https://direct.mit.edu/neco/article-abstract/16/10/2197/6863/Learning-Eigenfunctions-Links-Spectral-Embedding?redirectedFrom=fulltext
>
> [2] https://aaai.org/papers/11732-autoencoder-by-forest/
>
> [3] https://jmlr.org/papers/v9/biau08a.html
>
> [4] https://hal.science/hal-05006431v1/file/main.pdf
>
> [5] https://jmlr.org/papers/v7/meinshausen06a.html
>
> [6] https://projecteuclid.org/journals/annals-of-statistics/volume-47/issue-2/Generalized-random-forests/10.1214/18-AOS1709.full
>
> [7] https://proceedings.mlr.press/v32/denil14.html

---

> > ### Author Response · Authors · 2025-08-05
> > **Deadline approaching**
> >
> > Thanks again to the reviewers for their time and consideration. Please let us know if there any further points in need of clarification. If our rebuttals have resolved all outstanding issues, then by all means feel free to revise scores upward :)

---

> > ### Comment · Reviewer_58sa · 2025-08-06
> >
> > I thank the authors for their answers and clarifications. I have two remaining questions:
> >
> > - **Out-of-sample capabilities**: How can a new data point be encoded into the latent space using a trained Random Forest without having access to the original dataset?
> > - **PSD Proof**: The proof of the kernel's positive semidefinite property comes from its representation via a scaled inner product. Could the authors please provide a brief derivation showing the equivalence of Eq. (1) and the formula on line 182? It is difficult to see how these are equivalent, as the denominator in Eq. (1) depends on the first argument $x$, but the inner product representation on line 182 does not appear to have such a denominator.

---

> > > ### Author Response · Authors · 2025-08-06
> > > **Reply to 58sa**
> > >
> > > Thanks for your questions. Please find our answers below.
> > >
> > > -Test embeddings require the *leaf assignments* of the training data, not the original data itself. These assignments are stored in a (very) sparse binary matrix $\mathbf \Pi \in \\{0,1\\}^{n \times d_\Phi}$, with one column per leaf of the forest. From this, we calculate the inverse leaf sample sizes $\mathbf s$, which will be given by the reciprocals of the column sums of $\mathbf \Pi$. These are sufficient to compute our training kernel $\mathbf K \in [0,1]^{n \times n}$ via $B\mathbf K = \mathbf \Pi \mathbf S \mathbf \Pi^{\top}$, where $\mathbf S = \text{diag}(\mathbf s)$. We emphasize that $\mathbf \Pi$ is *very* light on memory relative to the true training data $\mathbf X \in \mathbb R^{d_X}$, given the extreme sparsity of the leaf assignment matrix. Let $\mathbf V, \mathbf \lambda$ denote the top $d_Z$ eigenvectors and eigenvalues (respectively) of the kernel matrix $\mathbf K$, with training embeddings given by $\mathbf Z = \sqrt{n} \mathbf V \mathbf \Lambda$, with $\mathbf \Lambda = \text{diag}(\mathbf \lambda)$. Now say we have a test point $\mathbf x_0 \in \mathbb R^{d_X}$. We run it through the RF to find its leaf assignments $\mathbf \psi \in \\{0,1\\}^{d_\Phi}$. The corresponding length-$n$ kernel vector is given by $\mathbf k_0 := \mathbf \Pi \mathbf S \mathbf \psi^{\top} = \\{k(\mathbf x_0, \mathbf x_i)\\}_{i=1}^n$. Now test embeddings are calculated via the Nyström formula: $\mathbf z_0 = \mathbf k_0 \mathbf Z \mathbf \Lambda^{-1}$. In this way, we can map test points to the embedding space using only learned parameters and kernel evaluations computable with a single pass of the RF. On our understanding, a *true out-of-sample encoding method* is one that can embed unseen points into a learned space without retraining or using the original training data. We hope R3 will agree that our method meets these criteria.
> > >
> > > -We are asked to show the equivalence of two alternative definitions of the RF kernel, one given by Eq. 1 and the other by a scaled inner product. Note that we define the canonical feature map as $\phi(\mathbf{x}) = \left[ \phi^{(1)}(\mathbf{x}), \ldots, \phi^{(B)}(\mathbf{x}) \right]$, with each tree-wise feature map given by $\phi^{(b)}(\mathbf{x}) = \pi^{(b)}(\mathbf{x}) \odot \sqrt{\mathbf{s}^{(b)}}$,
> > > in which $\pi^{(b)}(\mathbf{x}) \in \\{0,1\\}^{L_b}$ is a one-hot vector indicating the leaf assignment of $\mathbf{x}$ in tree $b$, where $L_b$ is the number of leaves in tree $b$;
> > > $\mathbf{s}^{(b)} \in \mathbb{R}^{L_b}$ is a vector of inverse leaf sample sizes, with entries $s_\ell^{(b)} = 1/ \sum_{i=1}^n \pi_\ell^{(b)}(\mathbf x_i)$;
> > > and $\odot$ denotes the Hadamard (element-wise) product. Now we expand the inner product:
> > > $$
> > > \langle \phi(\mathbf{x}), \phi(\mathbf{x}') \rangle
> > > = \sum_{b=1}^B \langle \phi^{(b)}(\mathbf{x}), \phi^{(b)}(\mathbf{x}') \rangle \nonumber \\
> > > = \sum_{b=1}^B \left\langle \pi^{(b)}(\mathbf{x}) \odot \sqrt{\mathbf{s}^{(b)}},  \pi^{(b)}(\mathbf{x}') \odot \sqrt{\mathbf{s}^{(b)}} \right\rangle \nonumber \\
> > > = \sum_{b=1}^B \sum_{\ell=1}^{L_b} \pi_\ell^{(b)}(\mathbf{x}) \pi_\ell^{(b)}(\mathbf{x}') s_\ell^{(b)}.
> > > $$
> > >
> > > If both $\mathbf{x}$ and $\mathbf{x}'$ fall in leaf $\ell$ of tree $b$, then $\pi_\ell^{(b)}(\mathbf{x}) \pi_\ell^{(b)}(\mathbf{x}') = 1$; otherwise, the product evaluates to $0$.
> > > Let $s_{\mathrm{leaf}(\mathbf{x})}^{(b)}$ denote the inverse sample size of the leaf containing $\mathbf{x}$ in tree $b$. Then $\langle \phi^{(b)}(\mathbf{x}), \phi^{(b)}(\mathbf{x}') \rangle = s_{\mathrm{leaf}(\mathbf{x})}^{(b)}$ if $\mathbf{x}, \mathbf{x}'$ fall in the same leaf of tree $b$, or 0 otherwise. Note that we have:
> > > $$
> > > s_{\mathrm{leaf}(\mathbf{x})}^{(b)} = \frac{1}{\sum_{i=1}^n k^{(b)}(\mathbf{x}, \mathbf{x}_i)},
> > > $$
> > >
> > > since the number of training points in the same leaf as $\mathbf{x}$ is given by $\sum_{i=1}^n k^{(b)}(\mathbf{x}, \mathbf{x}_i).$
> > >
> > > Substituting and dividing by $B$ gives:
> > >
> > > \begin{align}
> > > \frac{1}{B} \langle \phi(\mathbf{x}), \phi(\mathbf{x}') \rangle
> > > &= \frac{1}{B} \sum_{b=1}^B \frac{k^{(b)}(\mathbf{x}, \mathbf{x}')}{\sum_{i=1}^n k^{(b)}(\mathbf{x}, \mathbf{x}_i)},
> > > \end{align}
> > > which completes the derivation. We will be sure to rewrite the proof of Thm. 3.4 to make this reasoning more explicit.
> > >
> > > Thanks again to 58sa for their time and consideration. Please let us know if there are any further questions or points we can help to clarify.

---

> > > > ### Comment · Reviewer_58sa · 2025-08-08
> > > >
> > > > Thank you for your detailed response. All my concerns have been addressed, and I will increase my score accordingly.

---

### Note · Authors · 2025-08-15

We reiterate our thanks to all three reviewers for their valued feedback. Below we summarize our key clarifications from the review process and make our final remarks.

• Reviewer R1 (B5CF) was concerned our pipeline is complex due to multiple decoding options and hyperparameters. We clarify that RF models are famously robust with minimal tuning; all experiments used heuristic settings without grid search, yielding strong results.

• Reviewer R2 (FErW) noted RFAE is computationally demanding. While we acknowledge our runtime is nontrivial, no known method for this task is both faster and more accurate. They also found Figure 4 unclear in showing RFAE’s superiority; we have moved Table 2 (Appendix B) to the main text to show overall rankings. R2 also requested runtime data, which we provided in the rebuttal and will include in supplementary material.

• Reviewer R3 (58sa) argued that autoencoding with trees has been done before and is akin to KPCA. We acknowledge the conceptual overlap but stress that our RF kernel theory contributions are entirely novel, and prior works differ fundamentally in methodology. They noted our “one hyperparameter” claim was oversimplified; we fully agree and have updated the manuscript accordingly. R3 also asked whether decoding requires the training dataset; we clarify that it requires only leaf assignments and the RF, preserving out-of-sample mapping. Lastly, they raised questions about the proofs (Eq. 1, Thm. 3.4), which we clarified in the rebuttal and updated in the manuscript to make the reasoning more explicit.

Summary:
We present an autoencoder based on the random forest kernel, enabling effective dimensionality reduction and denoising for mixed tabular data. Given the largely positive initial reviews and stimulating discussion during revisions, we hope the reviewers will agree that our work makes a novel theoretical contributions and implements practical, high-performance tools for diverse tabular data processing tasks. We again thank the reviewers and AC for their thoughtful input and consideration.

---

### Decision · Program_Chairs · 2025-09-17

**Decision:**

Accept (poster)

**Comment:**

The paper proposes an autoencoder based on kernel PCA, using a kernel derived from a trained random forest.

The paper proposes an interesting approach to an important problem and also provides theoretical guarantees. The reviews are positive, and although there were initially a number of concerns, these have largely been addressed by the authors in discussion. I recommend publication.

The authors have promised a number of changes during discussion, and these should be included with care, especially the clarifications in the proof of theorem 3.4.